EMBO
Molecular Medicine

# 3,4-Dimethoxychalcone induces autophagy through activation of the transcription factors TFE3 and TFEB

Guo Chen[1,2,3,4,5,6], Wei Xie[1,2,3,4,5,6,7], Jihoon Nah[8], Allan Sauvat[1,2,3,4,5,6], Peng Liu[1,2,3,4,5,6] (iD),
Federico Pietrocola[1,2,3,4,5,6], Valentina Sica[1,2,3,4,5,6], Didac Carmona-Gutierrez[9,10],
Andreas Zimmermann[9,10] (iD), Tobias Pendl[9], Jelena Tadic[9], Martina Bergmann[9],
Sebastian J Hofer[9,10] (iD), Lana Domuz[9,11], Sylvie Lachkar[1,2,3,4,5,6], Maria Markaki[12],
Nektarios Tavernarakis[12,13] (iD), Junichi Sadoshima[8], Frank Madeo[9,10], Oliver Kepp[1,2,3,4,5,6,†,*] (iD) &
Guido Kroemer[1,2,3,4,5,6,14,15,16,†,**] (iD)

## Abstract

Caloric restriction mimetics (CRMs) are natural or synthetic compounds that mimic the health-promoting and longevity-extending effects of caloric restriction. CRMs provoke the deacetylation of cellular proteins coupled to an increase in autophagic flux in the absence of toxicity. Here, we report the identification of a novel candidate CRM, namely 3,4-dimethoxychalcone (3,4-DC), among a library of polyphenols. When added to several different human cell lines, 3,4-DC induced the deacetylation of cytoplasmic proteins and stimulated autophagic flux. At difference with other well-characterized CRMs, 3,4-DC, however, required transcription factor EB (TFEB)- and E3 (TFE3)-dependent gene transcription and mRNA translation to trigger autophagy. 3,4-DC stimulated the translocation of TFEB and TFE3 into nuclei both *in vitro* and *in vivo*, in hepatocytes and cardiomyocytes. 3,4-DC induced autophagy *in vitro* and in mouse organs, mediated autophagy-dependent cardioprotective effects, and improved the efficacy of anticancer chemotherapy *in vivo*. Altogether, our results suggest that 3,4-DC is a novel CRM with a previously unrecognized mode of action.

**Keywords** caloric restriction; caloric restriction mimetic; cardioprotection; TFE3; TFEB
**Subject Categories** Cancer; Cardiovascular System; Pharmacology & Drug Discovery

## Introduction

Caloric restriction (CR) is the first health span-extending and longevity-extending intervention that has been broadly characterized for its beneficial effects on many different species including yeast (*Saccharomyces cerevisiae*), nematodes (*Caenorhabditis elegans*), flies (*Drosophila melanogaster*), rodents (*Mus musculus*), and non-human primates (*Macaca mulatta*; Steinkraus *et al*, 2008; Mattison *et al*, 2012; Mercken *et al*, 2014; Hartl, 2016). Direct genetic interventions to inhibit autophagy prevent lifespan extension by CR in

1   Gustave Roussy Cancer Campus, Villejuif, France
2   INSERM, UMR1138, Centre de Recherche des Cordeliers, Paris, France
3   Equipe 11 labellisée par la Ligue Nationale contre le Cancer, Centre de Recherche des Cordeliers, Paris, France
4   Université de Paris, Paris, France
5   Metabolomics and Cell Biology Platforms, Gustave Roussy Cancer Campus, Villejuif, France
6   Sorbonne Université, Paris, France
7   Faculté de Médecine, Université Paris-Saclay, Kremlin-Bicêtre, France
8   Department of Cell Biology and Molecular Medicine, Rutgers-New Jersey Medical School, Newark, NJ, USA
9   LInstitute of Molecular Biosciences, University of Graz, Graz, Austria
10  BioTechMed-Graz, Graz, Austria
11  Department of Biology and Ecology, Faculty of Sciences, University of Novi Sad, Novi Sad, Serbia
12  Institute of Molecular Biology and Biotechnology, Foundation for Research and Technology-Hellas, Heraklion, Greece
13  Medical School, University of Crete, Heraklion, Greece
14  Pôle de Biologie, Hôpital Européen Georges Pompidou, AP-HP, Paris, France
15  Suzhou Institute for Systems Medicine, Chinese Academy of Sciences, Suzhou, China
16  Department of Women's and Children's Health, Karolinska Institute, Karolinska University Hospital, Stockholm, Sweden
    *Corresponding author. Tel: +33 0 1421 14516; E-mail: captain.olsen@gmail.com
    **Corresponding author. Tel: +33 0 1442 77667; E-mail: kroemer@orange.fr
    †These authors contributed equally to this work as senior authors

yeast and nematodes (Hansen *et al*, 2008; Eisenberg *et al*, 2009). Conversely, genetic induction of autophagy by removal of the autophagy inhibitor mechanistic target of rapamycin (mTOR) can induce a longevity phenotype in nematodes and flies (Vellai *et al*, 2003; Kapahi *et al*, 2004; de Cabo *et al*, 2014). Moreover, transgenic over-expression of the pro-autophagic gene *ATG5* and gain-of-function knock-in mutation in another pro-autophagic gene, *Beclin 1*, are sufficient to extend lifespan in mice (Pyo *et al*, 2013; Fernandez *et al*, 2018). These findings underscore the cause–effect relationship between CR-induced autophagy and its anti-aging effects (Madeo *et al*, 2015).

Given the logistic (and psycho-social) difficulties to maintain long-term CR, the concept of "CR mimetics" (CRMs) has been developed (Eisenberg *et al*, 2014; Madeo *et al*, 2014; Marino *et al*, 2014b). CR induces autophagy secondary to the depletion of acetyl coenzyme A (AcCoA), leading to the subsequent deacetylation of cellular proteins, mostly in the cytoplasm (because acetyltransferases have a low affinity for AcCoA, meaning that even minor reductions in AcCoA concentration can cause the inhibition of the enzymes), and induction of autophagy (Marino *et al*, 2014a; Pietrocola *et al*, 2015a). CRMs induce a similar cascade of events culminating in protein deacetylation and autophagy induction (Eisenberg *et al*, 2014; Madeo *et al*, 2014). One particular CRM, spermidine, which inhibits the autophagy-suppressive acetyltransferase EP300 (Pietrocola *et al*, 2015b), has been found to increase the life expectancy of yeast, nematodes, flies, and mice (Eisenberg *et al*, 2009). Moreover, nutritional spermidine uptake has been correlated with human longevity (Eisenberg *et al*, 2016; Kiechl *et al*, 2018). Another CRM, resveratrol, which activates the deacetylase sirtuin-1 (Howitz *et al*, 2003), has been shown to increase the lifespan of nematodes in an autophagy-dependent fashion (Morselli *et al*, 2010) and exhibits health-promoting effects, especially in overfed mice (Baur *et al*, 2006). Resveratrol also has anti-obesity and anti-diabetic effects in rodents (Baur *et al*, 2006). In mice, CRMs such as spermidine and resveratrol have broad cardioprotective effects (Stewart *et al*, 1990; Eisenberg *et al*, 2016) and enhance the efficacy of anticancer chemotherapies by stimulating immune responses against tumor-associated antigens (Walti *et al*, 1986; Pietrocola *et al*, 2016) echoing the fact that high spermidine uptake correlates with reduced cardiovascular and cancer-related mortality in human epidemiological studies (Kiechl *et al*, 2018; Pietrocola *et al*, 2018).

Based on these considerations, we engaged in the search for additional CRMs. Here, we report the identification of a novel candidate CRM among a collection of polyphenols. This agent, 3,4-dimethoxychalcone (3,4-DC), was subjected to an in-depth mechanistic analysis. Indeed, we found that 3,4-DC acts through the activation of pro-autophagic transcription factors and exhibits cardioprotective and potential anticancer effects in mice.

## Results

### Identification of 3,4-dimethoxychalcone (3,4-DC) as a candidate CRM

In an attempt to identify new autophagy inducers acting as CRMs, we initially screened a library of 200 polyphenols and polyamines (Appendix Table S1) for their capacity to induce the formation of cytoplasmic GFP-LC3 dots in human osteosarcoma U2OS cells (Fig 1A). Remarkably, among the eight compounds having the strongest effect at 30 μM five fell into the category of chalcones. For this reason, we re-screened all chalcones contained in the library again for their capacity to induce GFP-LC3 puncta and to cause cytoplasmic protein deacetylation (as determined by immunofluorescence staining) in U2OS (Fig 1B and C) and human H4 neuroglioma cells (Fig 1D and E). When the absence of effects on cellular viability was included as a criterion for bona-fide CRM effects, 3,4-dimethoxy chalcone (3,4-DC) stood out as the agent that caused the best combination of enhanced GFP-LC3 dots, reduced protein acetylation and absence of toxicity, both in U2OS cells (Fig 1F and G; Appendix Table S2) and in H4 cells (Fig 1H and I). Moreover, we used several agents known to increase the cytosolic acetyl coenzyme A (AcCoA) levels, namely dichloroacetate (DCA), which is an inhibitor of pyruvate dehydrogenase kinase, and two branched amino acids L-leucine (Leu) and α-ketoisocaproic acid (KIC), which undergo reductive carboxylation to yield AcCoA (Marino *et al*, 2014a). These three agents (Leu, KIC, and DCA) effectively inhibited 3,4-DC-induced deacetylation of cytoplasmic proteins as they suppressed 3,4-DC-induced autophagy (Fig 1J and K). Of, note 3,4-DC had no cytotoxic effects, as determined by vital staining with propidium iodide (Appendix Fig S1A and B).

---

**Figure 1.   Identification of 3,4-dimethoxychalcone (3,4-DC) as a potential caloric restriction mimetic (CRM).**

A   Human osteosarcoma U2OS-GFP-LC3 cells were treated with a library of polyphenols and polyamines for 6 h (compounds are listed in Appendix Table S1). And then, GFP-LC3 dots were counted to measure autophagy activity. Top 20 hits were shown in the right frame. Data are means ± SD of four replicates.

B–I   Human osteosarcoma U2OS cells (B, F) or neuroglioma H4 cells (D, H) stably expressing GFP-LC3 were treated with a library of chalcones (30 μM) as indicated or rapamycin (10 μM), or were left untreated for 6 h. The cells were then fixed, and GFP-LC3 dots were counted as an indicator for autophagy. Data are means ± SD of four replicates (\*$P < 0.05$; \*\*$P < 0.01$; \*\*\*$P < 0.001$; Student's *t*-test). Representative images are shown in (B, D). Scale bar equals 10 μm. (C, E) Acetylation: U2OS and H4 cells were treated as described above, followed by incubation with specific antibodies to block acetylated tubulin. Thereafter, immunofluorescence was conducted with antibodies against acetylated lysine residues and appropriate Alexa Fluor-conjugated secondary antibodies before the assessment of cytoplasmic fluorescence intensities. Representative images of acetylation are shown in (C, E). Scale bar equals 10 μm. (G, I) Viability: U2OS and H4 cells were treated with 30 μM of the indicated chalcones for 24 h, and then, nuclei were stained with Hoechst 33342 and the number of cells harboring normal nuclei (i.e., non-pyknotic, "healthy cells") was determined. All chalcones were hierarchically clustered upon z-scoring the following phenotypes: autophagy (number of GFP-LC3 dots), viability (number of healthy cells), and acetylation (cytoplasmic fluorescence intensity of acetylated lysine residues) in U2OS and H4 cells. Results are reported as a heatmap, and abbreviations are listed in Appendix Table S2.

J, K   U2OS cells were treated with 3,4-DC in the presence or absence of dichloroacetate (DCA), α-ketoisocaproic acid (KIC), or L-leucine (Leu). GFP-LC3 dots were quantified in (J). Acetylation staining was performed as previously described, and acetylation intensity in the cytoplasm was measured in (K). Data are means ± SD of six replicates (\*\*\*$P < 0.001$ vs. DMSO/Ctr; $^{\#\#}P < 0.01$, $^{\#\#\#}P < 0.001$ vs. 3,4-DC/Ctr; Student's *t*-test).

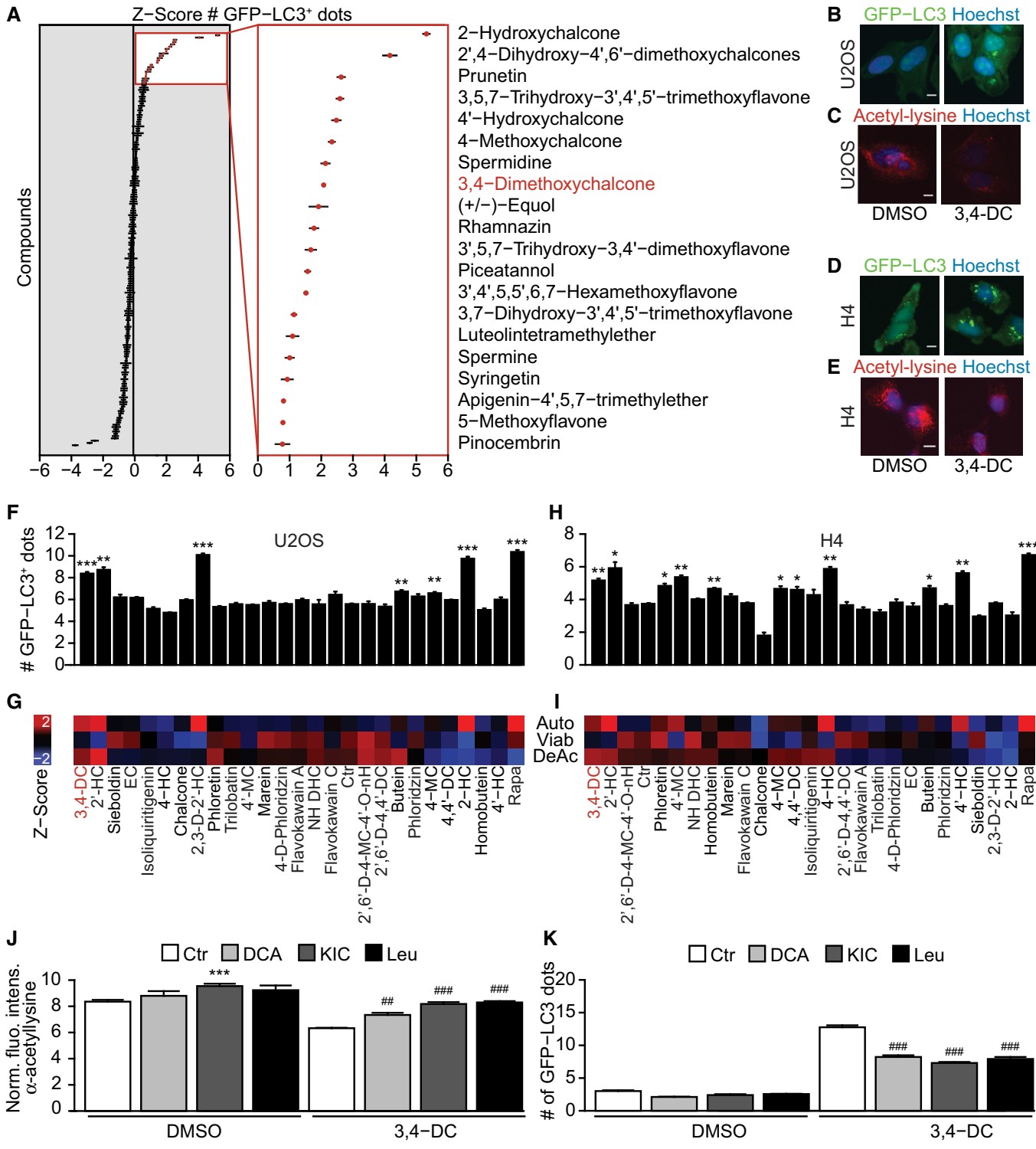

**Figure 1.**

## 3,4-DC induces autophagic flux

3,4-DC caused a dose-dependent increase in LC3 lipidation (detectable as an increase in the electrophoretic mobility of LC3 yielding LC3-II) (Fig 2A and B). Time-course analyses revealed a transient reduction in the abundance of sequestosome 1 (SQSTM1, best known as p62), followed by its increase (Fig 2C and D). 3,4-DC also stimulated the deacetylation of histone H2A (Fig EV1A and B). In the presence of the lysosomal inhibitor chloroquine (CQ), the lipidation of LC3 was further increased (Fig 2E and F), commensurate

with the redistribution of GFP-LC3 into dots (Fig 2G and H). 3,4-DC caused a reduction in the abundance of plasmid-encoded p62 fused to a hemagglutinin (HA) tag (the expression of which was under the control of the cytomegalovirus promoter and hence largely independent of transcriptional host cell regulation), and this effect was blocked by chloroquine (Fig EV1C and D). These results obtained in human hepatoma HepG2 cells (Fig 2A–F) could be recapitulated in H4 cells (Fig EV1E, F and H–J) and U2OS cells (Fig EV1G). In PC12 neuronal cells expressing a doxycycline (Dox)-inducible polyglutamine-74 (Q74)-tagged GFP, 3,4-DC also reduced the abundance of this autophagic substrate to the same level as the positive control, torin1, which is an inhibitor of mTOR (Appendix Fig S2A and B).

In cells engineered to express a tandem GFP-RFP-LC3 fusion protein (Kimura *et al*, 2007), 3,4-DC could increase the abundance of autophagosomes (RFP and GFP fluorescence) and autolysosomes (RFP fluorescence only), contrasting with autophagic flux inhibitors such as chloroquine and bafilomycin A1 (BafA1), which interfere with the formation of autophagosomes (Fig 2I and J). These effects of 3,4-DC were accompanied by a dose- and time-dependent inhibition of mTOR (as indicated by reduced phosphorylation of the mTOR substrate p70S6K) (Fig 3A and B). The 3,4-DC-induced formation of LC3-II was lost in $ATG5^{-/-}$ cells, which however continued to manifest the upregulation of p62 and LC3-I (Fig EV2A–C, Appendix Fig S3A and B). The upregulation of p62 that could be observed after a prolonged (16 h) exposure to 3,4-DC was lost upon preincubation of the cells with cycloheximide (CHX), an inhibitor of protein synthesis (Fig EV2D and E, Appendix Fig S3C). Moreover, CHX and the RNA synthesis inhibitor actinomycin D (AMD) prevented the induction of GFP-LC3 dots by 3,4-DC (Fig 3C and D). These observations, which were obtained in several cell lines (Figs 3 and EV2A–G, Appendix Fig S3A–C), plead in favor of the capacity of 3,4-DC to activate a pro-autophagic transcriptional program. Indeed, cells treated with 3,4-DC manifested an increase in the mRNA expression of the genes encoding ATG14, LAMP1, LC3B, P62, and ULK1 (Figs 3E and EV2H). CHX inhibited the 3,4-DC-induced increase in LAMP1 protein expression but not of *LAMP1* mRNA (Appendix Fig S4A–C). Of note, while torin1 induced autophagy even in cells exposed to CHX or cells that have been enucleated to generate cytoplasts, 3,4-DC-stimulated autophagy was suppressed by CHX and depended on the presence of nuclei within the cells (Figs EV2I, and 3F and G).

### Implication of TFEB and TFE3 in 3,4-DC-induced autophagy

Using U2OS cells stably expressing GFP fused to transcription factor EB (TFEB), we found that several chalcones including 3,4-DC were able to stimulate the translocation of GFP-TFEB from the cytoplasm to the nucleus (Fig 4A and B). Similar results were obtained when endogenous TFEB was visualized by immunofluorescence (Fig 4C and D) or detected by immunoblot after subcellular fractionation (Fig 4E). Of note, spermidine differed from torin1 and 3,4-DC in thus far that it was unable to stimulate the nuclear translocation of TFEB (Fig EV3A–C). Subcellular fractionation also led to the observation that TFE3, a protein that falls into the same family of transcription factors as TFEB (together with MiTF), translocated from the cytoplasm to the nuclei (Fig 4F), as confirmed by immunofluorescence staining (Fig EV3D and E). TFEB and TFE3 are known to favor lysosomal biogenesis (Settembre *et al*, 2011; Martina *et al*, 2014; Napolitano & Ballabio, 2016; Puertollano *et al*, 2018). Accordingly, 3,4-DC induced an increase in the expression of the lysosomal protein LAMP1 (Fig 3A and B, Appendix Fig S4B and C) and increased the lysosomal mass as indicated by a rise in the number of Lamp1-RFP or LysoTracker-positive dots per cell (Appendix Fig S5A–D) that occurred in a protein synthesis-dependent fashion (Appendix Fig S5E). As shown by immunofluorescence staining and subcellular fractionation followed by immunoblot, the 3,4-DC-induced TFEB translocation was inhibited by TSC2 knockout (Fig 4G–I), a manipulation that causes constitutive activation of mTOR (Zhang *et al*, 2003). The TSC2 knockout abolished the 3,4-DC-induced induction of p62 and LC3-II protein levels (Fig 4J–L). TFEB is phosphorylated at residues S142, S138, and S211 by mTOR to regulate its activity. Phosphorylation of TFEB at S211 enhances the binding of TFEB to 14-3-3 protein, thus favoring the retention of TFEB in the cytoplasm (Martina *et al*, 2012). Similar to torin, 3,4-DC caused the dephosphorylation of TFEB at S211 (Fig 4M and N), in line with its capacity to inhibit the phosphorylation of another mTOR substrate, P70S6K (see above). Thus, 3,4-DC induces TFEB activation through the inhibition of mTOR.

---

**Figure 2.  3,4-DC induced autophagic flux.**

A–D  Hepatoma HepG2 cells were treated with the indicated concentrations of 3,4-DC for 8 h (A, B) or with 30 μM 3,4-DC for the indicated time points (C, D). Then, cells were processed to measure LC3 and p62 protein levels by SDS–PAGE and immunoblot (A, C). GAPDH was measured as a loading control. Band intensities of p62, GAPDH, LC3-I, and LC3-II were measured, and ratios of p62 or LC3-II vs. GAPDH (LC3-II/GAPDH, p62/GAPDH) and LC3-II vs. LC3-I (LC3-II/LC3-I) were calculated in (B, D). Data are means ± SEM of at least three independent experiments (LC3-II/GAPDH: \*P < 0.05, \*\*P < 0.01; p62/GAPDH: #P < 0.05, ###P < 0.001; LC3-II/LC3-I: $P < 0.05; Student's *t*-test).

E, F  HepG2 were treated with 30 μM 3,4-DC for 8 h in the presence or absence of chloroquine (CQ, 50 μM) for 4 h. SDS–PAGE and immunoblot were performed as in (A). Band intensities of LC3-II and GAPDH were assessed, and their ratio (LC3-II/GAPDH) was calculated in (F). Data are means ± SEM of at least three independent experiments (\*P < 0.05, \*\*P < 0.01 vs. untreated control; ##P < 0.01 vs. CQ; Student's *t*-test).

G, H  U2OS-GFP-LC3 cells were treated with 30 μM 3,4-DC, 10 μM rapamycin, or left untreated for 16 h in the presence or absence of CQ for 4 h. GFP-LC3 dots were measured. Data are means ± SD of four replicates (\*\*\*P < 0.001 vs. untreated; ###P < 0.001 vs. CQ; Student's *t*-test).

I, J  U2OS cells stably expressing LC3 fused with tandem fluorescent GFP-RFP proteins (GFP-RFP-LC3) were treated with the indicated raising doses of 3,4-DC or 50 μM chloroquine (CQ) for 16 h. After fixation, GFP-LC3 and RFP-LC3 dots were measured by automated image acquisition and analysis. GFP and RFP double-positive LC3 dots indicating autophagosomes (GFP+) and GFP-negative but RFP-positive LC3 dots indicating autolysosomes (GFP−) were counted in (J). Data are means ± SD of four replicates (\*\*P < 0.01;\*\*\*P < 0.001; #P < 0.05; ##P < 0.01; ###P < 0.001; Student's *t*-test). Representative images are shown in (I).

Data information: Scale bar equals 10 μm. Immunoblots in (A, C, and E) were run together on the same gel, then blots were cut into horizontal stripes, and probed separately.
Source data are available online for this figure.

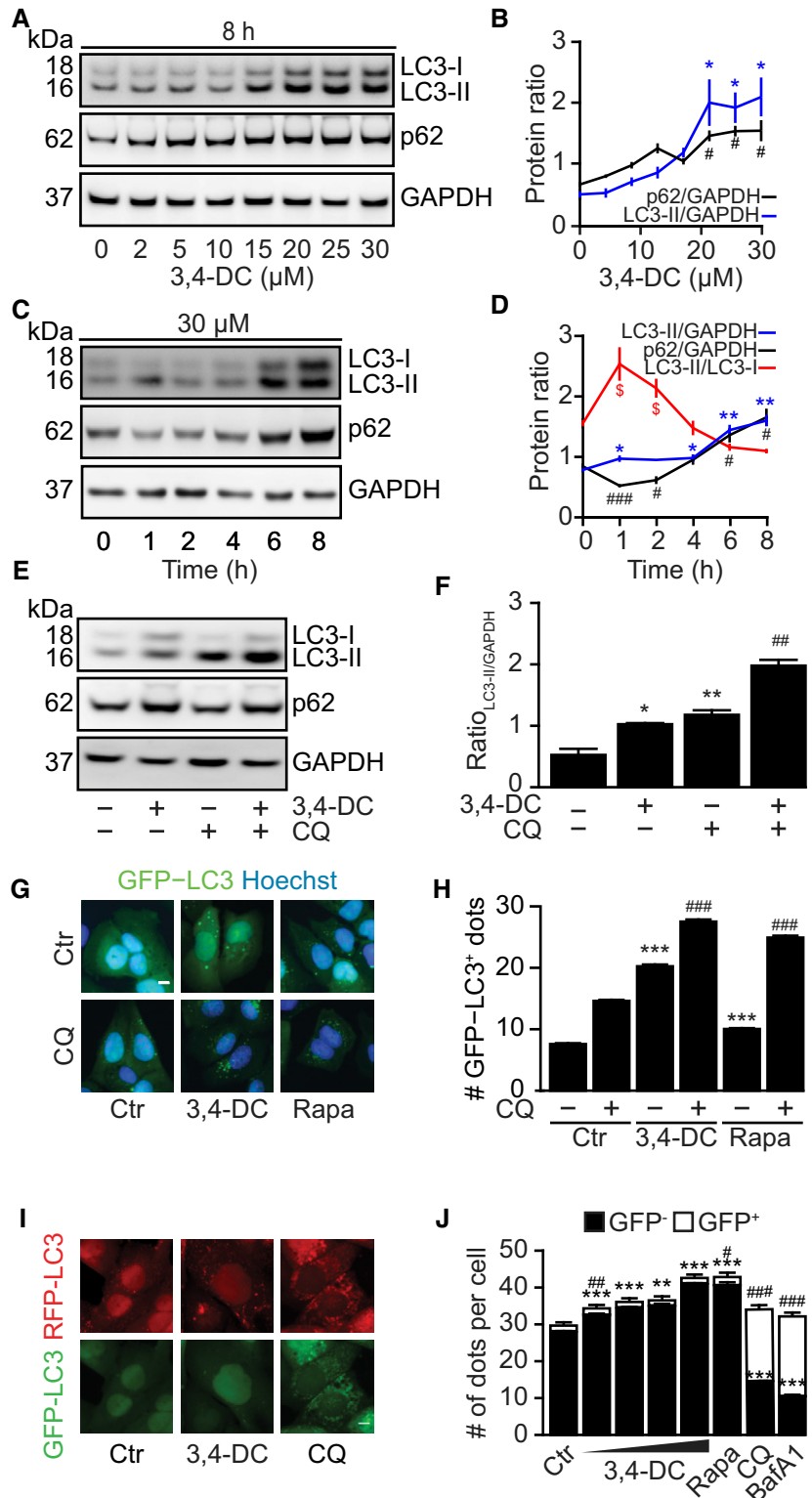

**Figure 2.**

Knockdown of TFEB with three different siRNAs (Fig 5A) partially reduced the formation of GFP-LC3 puncta in response to 3,4-DC (Fig 5B and C). Similarly, knockout of TFEB by CRISP/Cas9 technology (Fig 5D) partially reduced 3,4-DC-induced GFP-LC3

puncta (Fig 5E and F). Since these effects were incomplete, we also knocked out TFE3, alone or in combination with TFEB, to generate TF double knockout (TF DKO) cells (Fig 5G). The TF DKO was particularly efficient in reducing 3,4-DC-induced LC3 lipidation and

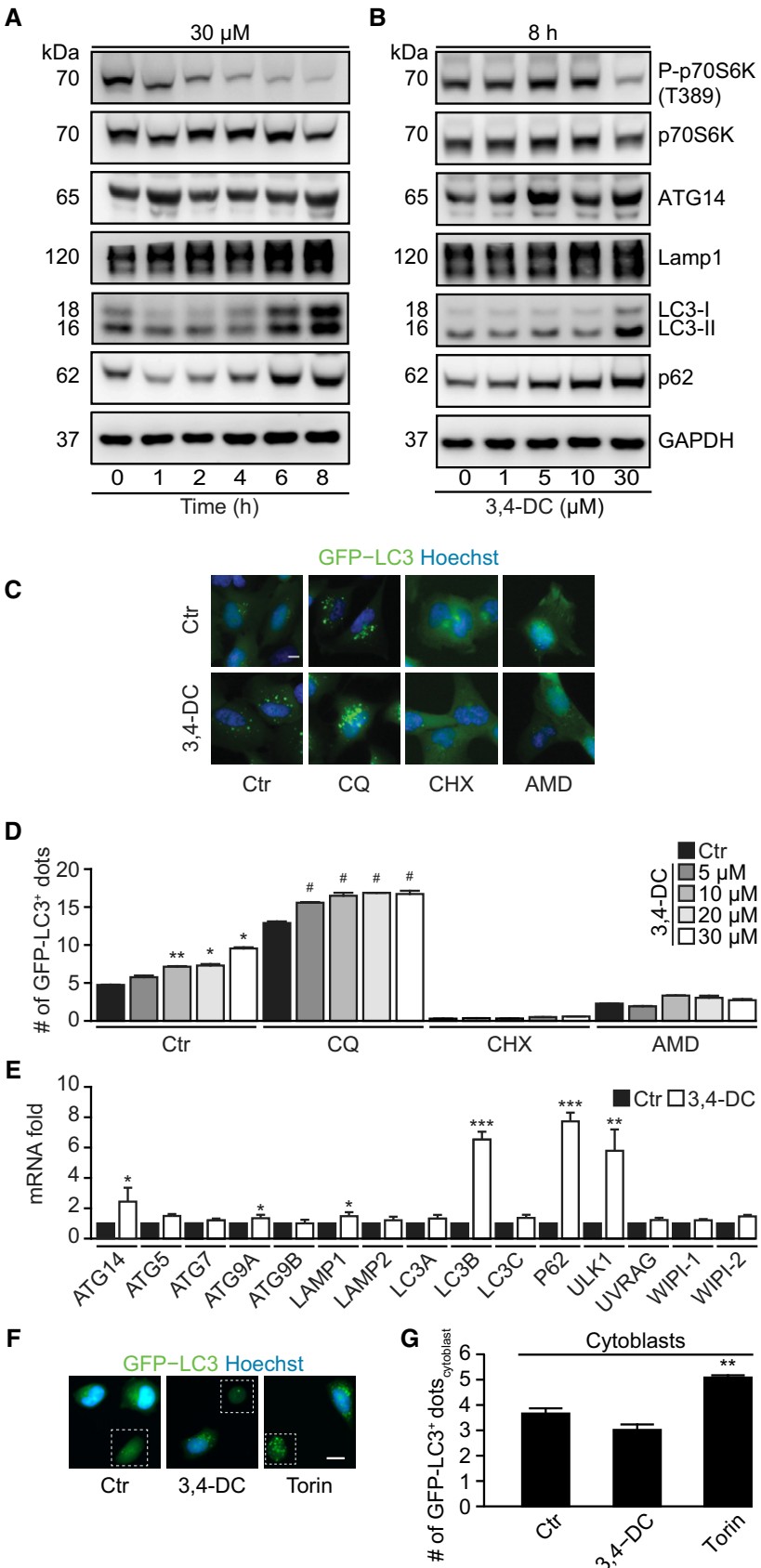

**Figure 3.**

**Figure 3.   3,4-DC induces autophagy in a transcription- or translation-dependent mechanism.**

A, B   HepG2 cells were treated for the indicated time with 30 μM 3,4-DC (A) or indicated dose of 3,4-DC for 8 h (B). Thereafter, cells were collected and SDS–PAGE and immunoblots were performed as described before. Total p70S6K, phosphorylated p70S6K at T389 (P-p70S6K), Atg14, Lamp1, LC3, p62, and GAPDH protein levels were measured with specific antibodies.

C, D   H4-GFP-LC3 cells were treated with 30 μM 3,4-DC in the presence or absence of CHX or AMD or with CQ for 16 h as controls, as indicated. GFP-LC3 dots were quantified in (D). Data are means ± SD of three replicates (*$P < 0.05$, **$P < 0.01$ vs. untreated control; #$P < 0.05$ vs. untreated with CQ; Student's $t$-test).

E   RNA expression levels 16 h after exposure to 3,4-DC of the indicated genes in HepG2 cells were measured by RT–PCR and normalized to the expression of the housekeeping gene (GAPDH). Data are means ± SD of three replicates (*$P < 0.05$; **$P < 0.01$; ***$P < 0.001$; Student's $t$-test).

F, G   U2OS-GFP-LC3 cells were enucleated to obtain cytoplasts, which were further treated with 3,4-DC and torin1. GFP-LC3 dots in cytoplasts were counted as shown in (G). Data are means ± SD of four replicates (**$P < 0.01$; Student's $t$-test). Representative images with cytoplast marked by a white square are shown in (F).

Data information: Scale bar equals 10 μm. Samples for immunoblots in (A and B) were run in parallel instances, then cut into horizontal stripes, and probed separately. Blots probed for P-p70S6K were stripped and reprobed to detect p70 independently from its phosphorylation status. The same applies for LAMP1 and GAPDH.
Source data are available online for this figure.

---

p62 induction (Fig 5G), as well as GFP-LC3 puncta (Fig 5H) and lysosomal biogenesis (Fig 5I). Similarly, the combination of the knockout of TFEB and the knockdown of TFE3 yielded a stronger inhibition of 3,4-DC-induced LC3 lipidation and p62 accumulation than deletion/depletion of each of these factors alone (Fig EV3F). In contrast, knockdown of MiTF failed to affect LC3 lipidation in TFEB knockout cells (Fig EV3G). In contrast to WT controls, TF DKO cells failed to manifest the 3,4-DC-elicited expression of key autophagy genes (Fig 5J). In sum, 3,4-DC stimulated the activation of TFEB and TFE3 that together account for autophagy induction.

We also compared the effects of 3,4-DC to another, structurally related compound, 4,4′-DC, that we recently described for its autophagy-inducing effects (Carmona-Gutierrez *et al*, 2019). While 3,4-DC causes classical autophagy (with autophagosomes in the cytoplasm), 4,4′-DC causes accumulation of GFP-LC3 dots in or around nuclei (Appendix Fig S6A–E). More importantly, the side-by-side comparison of both compounds revealed that 3,4-DC (but not 4,4′-D) induced autophagy through the activation of transcription factors of the TFE family (in particular TFEB and TFE3), while 4,4′-DC (but

not 3,4-DC) stimulated autophagy via the inhibition of GATA transcription factors (Fig EV4).

**3,4-DC induces autophagy-dependent cardioprotection**

The aforementioned data suggest that 3,4-DC induces autophagy *in vitro*, in cultured human or rodent cells. In the next step, we injected 3,4-DC intraperitoneally (i.p.) into C57Bl/6 mice at doses that did not induce any manifest signs of toxicity (weight loss) (Appendix Fig S7) and determined its *in vivo* effects. Subcellular fractionation of liver and heart tissues, followed by immunoblotting, revealed that 3,4-DC induced the nuclear translocation of TFEB and TFE3 (Fig 6A–D). When injected into mice ubiquitously expressing a GFP-LC3 transgene (Mizushima *et al*, 2004; Mizushima, 2009), 3,4-DC caused the induction of GFP-LC3 puncta in the cytoplasm of hepatocytes (Fig 6E and F) and cardiomyocytes (Fig 6I and J). This effect was more pronounced when the mice received leupeptin, an inhibitor of lysosomal proteases, 2 h before sacrifice (Esteban-Martinez & Boya, 2015), confirming that 3,4-DC increments autophagic flux

---

**Figure 4.   3,4-DC induced TFEB-dependent lysosomal biogenesis and autophagy.**

A, B   U2OS cells stably expressing GFP-TFEB fusion protein were treated with indicated chalcones at 30 μM for 6 h. GFP intensities in nuclei and cytoplasm were measured, and the ratio of GFP intensities in nuclei and cytoplasm was calculated to indicate TFEB translocation to nuclei (A). Data are means ± SD of four replicates (*$P < 0.05$;**$P < 0.01$;***$P < 0.001$; Student's $t$-test). Representative images are shown in (B). Scale bar equals 10 μm.

C, D   U2OS cells were treated with 30 μM 3,4-DC for 6 h, and then, endogenous TFEB translocation was assessed by immunostaining (C). TFEB intensities in nuclei and cytoplasm were measured, and the ratio of TFEB intensities in nuclei and cytoplasm was calculated to indicate TFEB translocation to nuclei (D). Data are means ± SD of four replicates (***$P < 0.001$; Student's $t$-test). Representative images are shown in (C). Scale bar equals 10 μm.

E, F   U2OS cells were treated with 30 μM 3,4-DC for 6 h or were left untreated. Cytoplasmic and nuclear fractions were assessed for nuclear translocation of the transcription factors TFEB and TFE3 by SDS–PAGE.

G–I   MEF wild-type and TSC2 knockout (TSC2 KO) cells were treated with 3,4-DC for 6 h. The cells were fixed, and immunofluorescence was performed to detect endogenous TFEB. The ratio of TFEB intensities between nuclei and cytoplasm was calculated in (H). Data are means ± SD of eight replicates (***$P < 0.001$ vs. Ctr; ###$P < 0.001$ vs. WT/3,4-DC; Student's $t$-test). Representative images are shown in (G). Scale bar equals 10 μm. (I) After the treatment, nuclei were isolated and Western blot was performed to detect nuclear and cytosolic TFEB protein levels.

J–L   MEF wild-type and TSC2 KO cells were collected and lysed for Western blot after the treatment with 3,4-DC. Antibodies against LC3, p62, TSC2, or GAPDH were administered to detect protein levels, and phosphorylation of mTOR substrate p70 S6K at threonine 389 (T389) was measured with the phosphorylation-specific antibody (P-p70(T389)) and anti-p70 antibody. The band intensities of LC3-II, p62, and GAPDH were measured with ImageJ, and the ratios of LC3-II/GAPDH and p62/GAPDH were calculated in (K) and (L). Data are means ± SEM of at least three independent experiments (**$P < 0.01$, ***$P < 0.001$ vs. Ctr; #$P < 0.05$, ##$P < 0.01$ vs. WT/3,4-DC; Student's $t$-test).

M, N   U2OS cells treated with 3,4-DC or torin for 6 h were collected and processed for Western blot. TFEB phosphorylation at serine 211 was checked with the phosphorylation-specific antibody (P-TFEB(S211)), and total TFEB protein level was also detected with anti-TFEB. The band intensities of P-TFEB and total TFEB were measured with ImageJ, and their ratio was calculated in (N). Data are means ± SEM of at least three independent experiments (*$P < 0.05$ vs. Ctr; Student's $t$-test).

Data information: Samples for immunoblots were run together in one gel (E, F) or in several parallel gels (I, J, M), then blots were cut into horizontal stripes, and probed separately. Blots probed for P-p70 were stripped and reprobed to detect p70 independently from its phosphorylation status.
Source data are available online for this figure.

---

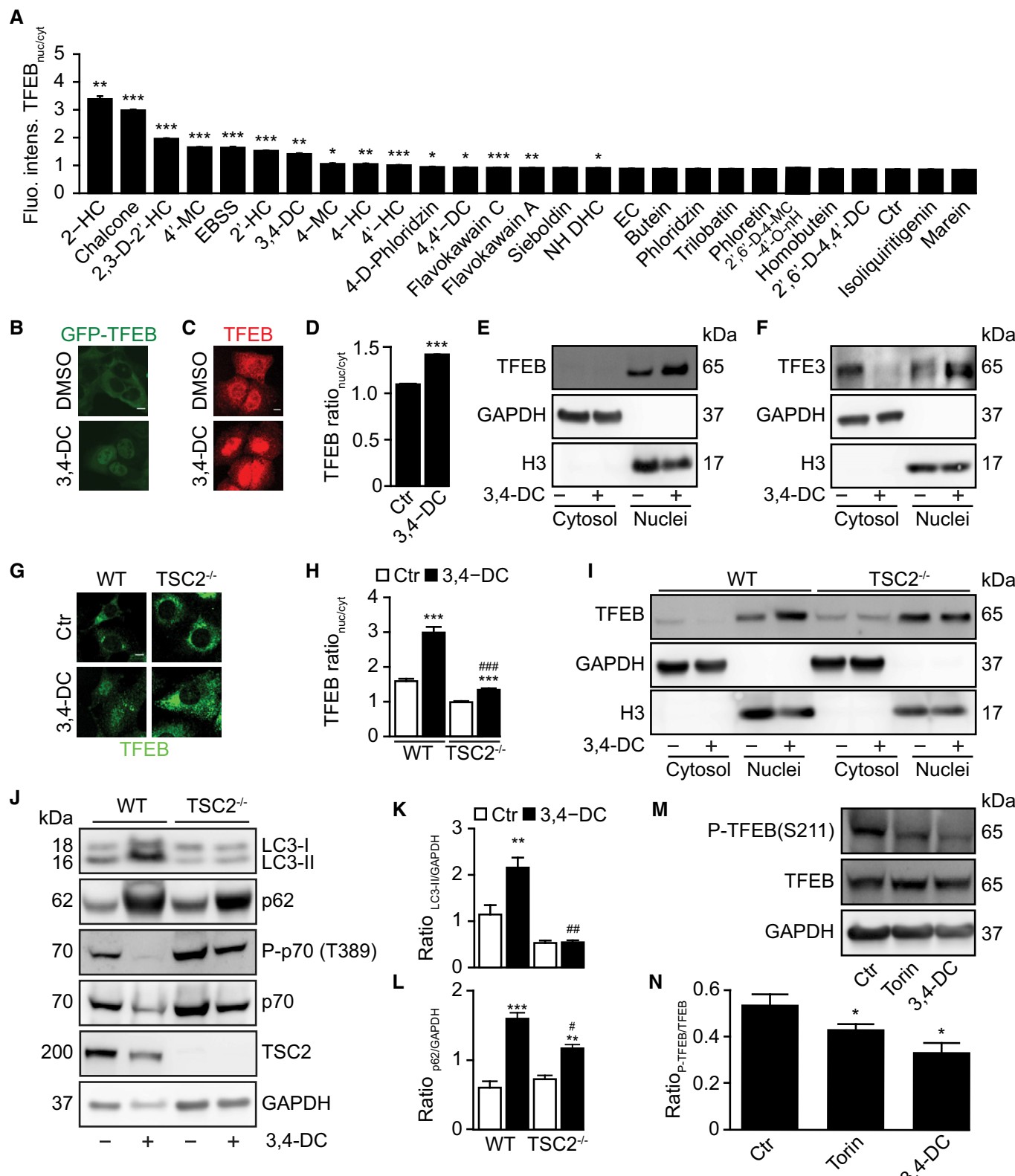

**Figure 4.**

(Fig 6E, F, I and J). Moreover, in non-transgenic C57Bl/6 mice, LC3 manifestly underwent lipidation, as indicated by an increase in the relative abundance of LC3-II both in the liver (Fig 6G and H) and in the heart (Fig 6K and L). Based on these results, we wondered whether 3,4-DC would induce cytoprotective autophagy. To clarify this issue, WT mice or animals bearing a cardiac muscle-specific *Atg7*

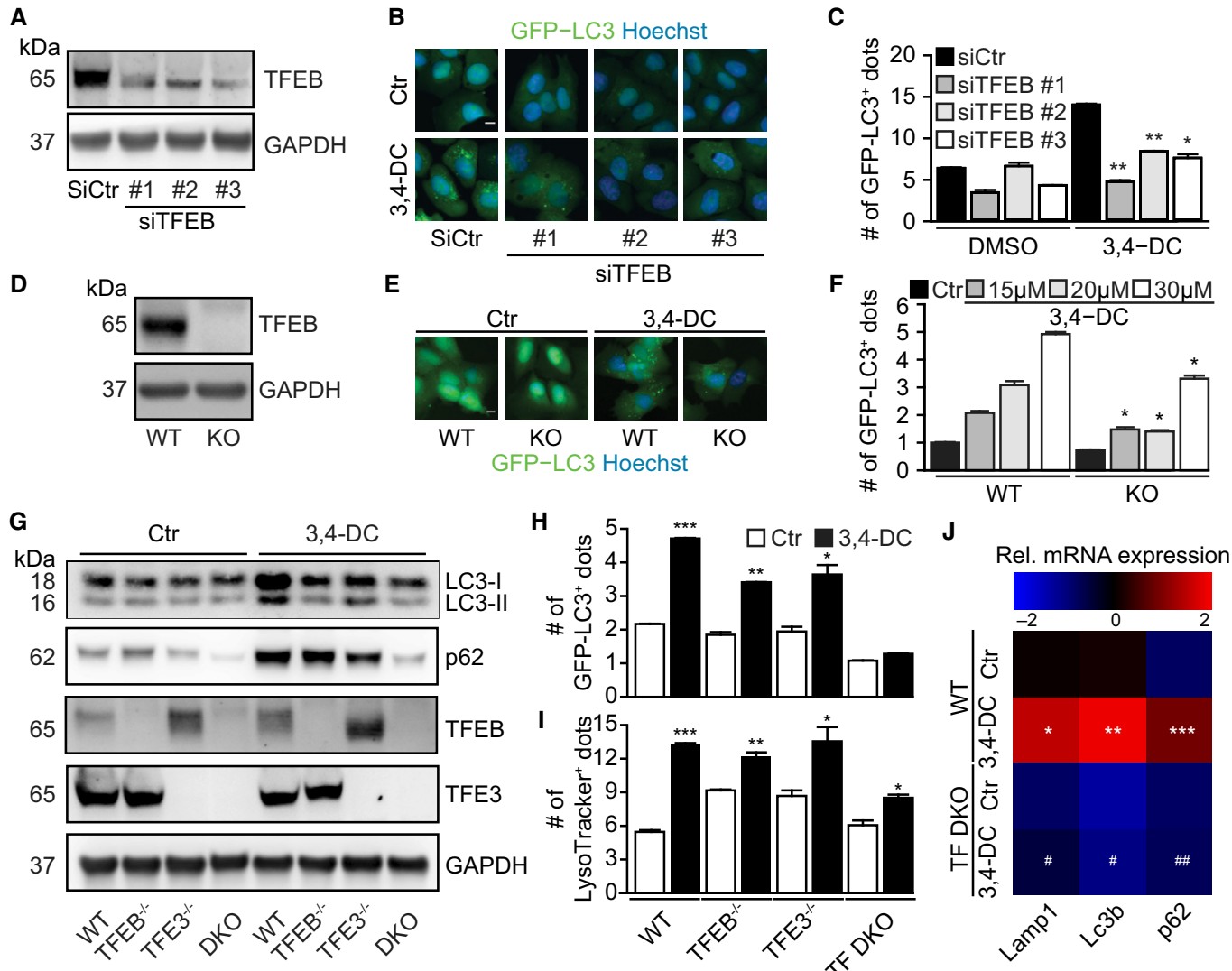

**Figure 5.  3,4-DC induced TFEB- and TFE3-dependent autophagy.**

A–F  U2OS-GFP-LC3 cells transfected with three individual siRNAs specifically targeting TFEB (siTFEB-#1, siTFEB-#2, and siTFEB-#3) or scrambled siRNA (siCtr) (A-C), or U2OS wild-type (wt) and TFEB knockout (TFEB KO) GFP-LC3 cells (D–F) were treated with 3,4-DC as indicated for 16 h. TFEB knockdown efficiency by TFEB siRNAs and TFEB deficiency by knockout were checked by SDS–PAGE and immunoblot (A, D). GFP-LC3 dots were quantified as indicator of autophagy (C, F). Data are means ± SD of three replicates (*$P$ < 0.05, **$P$ < 0.01 vs. siCtr or WT cells treated with 3,4-DC; Student's $t$-test). Representative images are shown in (B, E). Scale bar equals 10 μm.

G, H  U2OS wild-type (WT), TFEB or TFE3 knockout, or double knockout (TF DKO) cells were treated with 30 μM 3,4-DC for 16 h. Following this, cells were collected and SDS–PAGE and immunoblots were performed as described before. TFEB, TFE3, LC3, p62, and GAPDH protein levels were measured with specific antibodies (G). GFP-LC3 dots were quantified as indicator of autophagy (H). Data are means ± SD of three replicates (*$P$ < 0.05, **$P$ < 0.01, ***$P$ < 0.001 vs. Ctr, respectively; Student's $t$-test).

I  U2OS WT, knockout for TFEB (TFEB$^{-/-}$) or TFE3 (TFE3$^{-/-}$), or double knockout (DKO) cells were treated with 3,4-DC for 24 h and stained with LysoTracker Red for 30 min. Thereafter, the red (positive) dots were measured. Data are means ± SD of four replicates (*$P$ < 0.05; **$P$ < 0.01; ***$P$ < 0.001; Student's $t$-test).

J  Relative mRNA expression levels of key autophagy genes Lamp1, Lc3b, and p62 were detected in control and 3,4-DC-treated U2OS cells. Data are depicted as a heatmap showing means of at least three independent experiments (*$P$ < 0.05, **$P$ < 0.01, ***$P$ < 0.001 vs. WT/Ctr and #$P$ < 0.05, ##$P$ < 0.01, as compared to WT treated with 3,4-DC; Student's $t$-test).

Data information: Samples for immunoblots in (A and B) were run together on one gel and in (G) were run in parallel gels, then blotted, cut into horizontal stripes, and probed separately.

Source data are available online for this figure.

knockout (Atg7cKO) were treated with vehicle or were pretreated with 3,4-DC and then subjected to cardiac ischemia. 3,4 DC was able to reduce the relative volume of the myocardial infarction in WT, but not in Atg7cKO mice (Fig 7A–C), indicating that 3,4-DC can mediate cardioprotection through autophagy.

**3,4-DC improves chemotherapy-induced tumor growth reduction in an immune system-dependent fashion**

Autophagy inhibition may restrain or enhance tumor growth in a context-dependent fashion (Galluzzi *et al*, 2015). Established

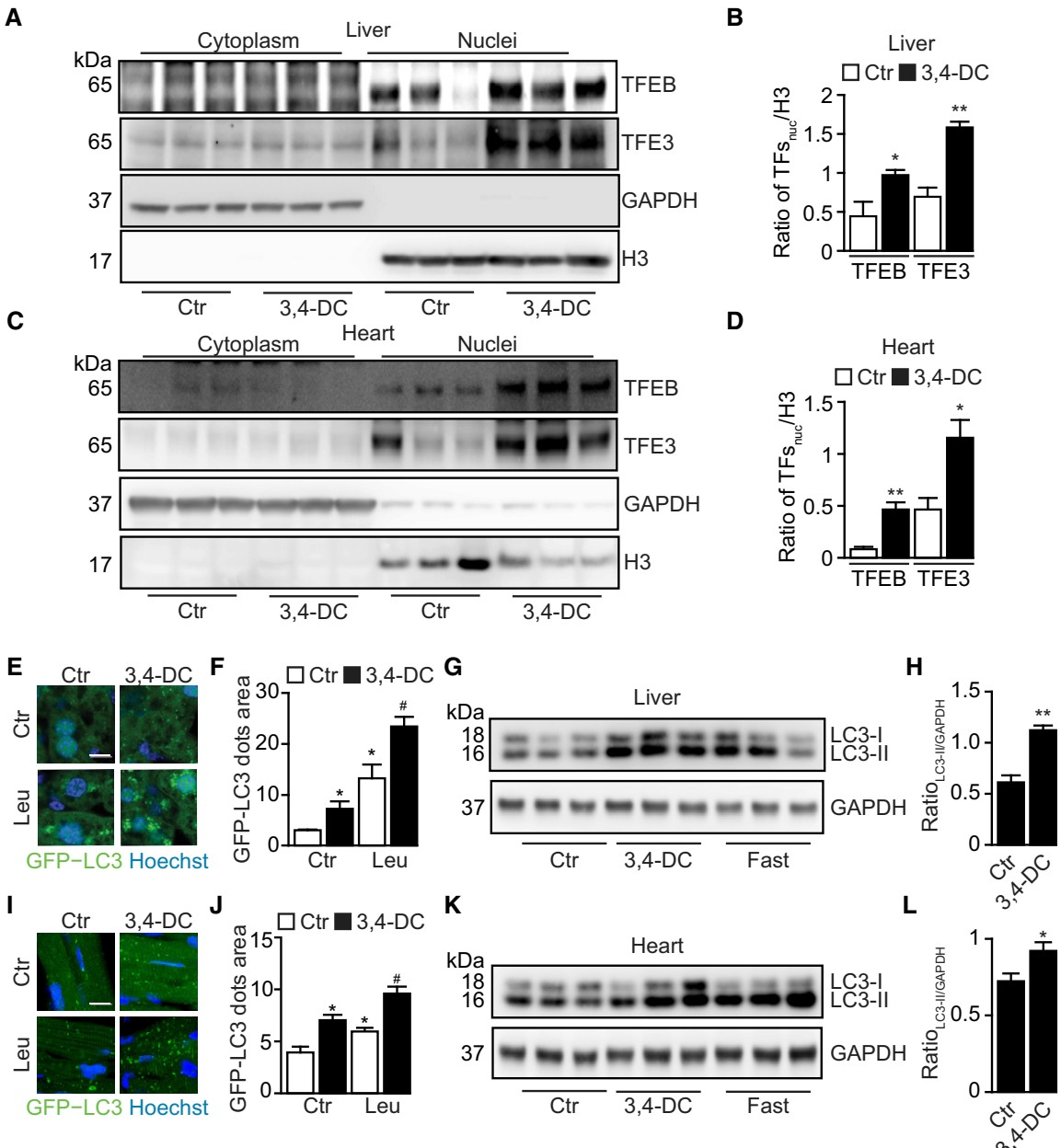

**Figure 6. 3,4-DC induces autophagy *in vivo*.**

A–D Female C57/BL6 animals were treated with 3,4-DC for 24 h. Following this, liver and heart tissues were excised and cells were subjected to subcellular fractionation. Thereafter, proteins were separated by SDS–PAGE and immunoblots were performed to detect the nuclear translocation of TFEB and TFE3 (A, C). GAPDH and H3 were used as controls for cytosolic and nuclear fractions, respectively. Band intensities of nuclear TFEB, TFE3, and H3 were measured, and ratios of nuclear TFs vs. H3 (TFs$_{nuc}$/H3) were calculated in (B, D). Data are means ± SEM of at least three mice (*$P < 0.05$, **$P < 0.01$; vs. Ctr; Student's $t$-test).

E, F GFP-LC3-expressing mice were i.p. injected with 3,4-DC for 24 h. Leupeptin (Leu) was used to test autophagic flux *in vivo*, and GFP-LC3 dots were measured in liver tissue. Data are means ± SEM of at least three mice (*$P < 0.05$ vs. Ctr without Leu; #$P < 0.05$ vs. Ctr with Leu; Student's $t$-test). Scale bar equals 10 μm.

G, H C57/BL6 mice were i.p. injected with 3,4-DC or were starved for 24 h. Following this, liver tissue was excised and subjected to SDS–PAGE and immunoblot (G). GAPDH was measured as a loading control. Band intensities of GAPDH and LC3-II were measured, and ratios LC3-II vs. GAPDH (LC3-II/GAPDH) were calculated in (H). Data are means ± SEM of at least three mice (**$P < 0.01$; Student's $t$-test).

I, J GFP-LC3-expressing mice were i.p. injected with 3,4-DC for 24 h. Leupeptin (Leu) was used to test autophagic flux *in vivo*, and GFP-LC3 dots were measured in heart tissue. Data are means ± SEM of at least three mice (*$P < 0.05$ vs. Ctr without Leu; #$P < 0.05$ vs. Ctr with Leu; Student's $t$-test). Scale bar equals 10 μm.

K, L C57/BL6 mice were i.p. injected with 3,4-DC or were starved for 24 h. Following this, heart tissue was excised and subjected to SDS–PAGE and immunoblot (K). GAPDH was measured as a loading control. Band intensities of GAPDH and LC3-II were measured, and ratios LC3-II vs. GAPDH (LC3-II/GAPDH) were calculated in (L). Data are means ± SEM of at least three mice (*$P < 0.05$; Student's $t$-test).

Data information: Samples for immunoblots were run together on the same gel (G, K) or on parallel gels (A, C), then blotted, cut into horizontal stripes, and probed separately. Source data are available online for this figure.

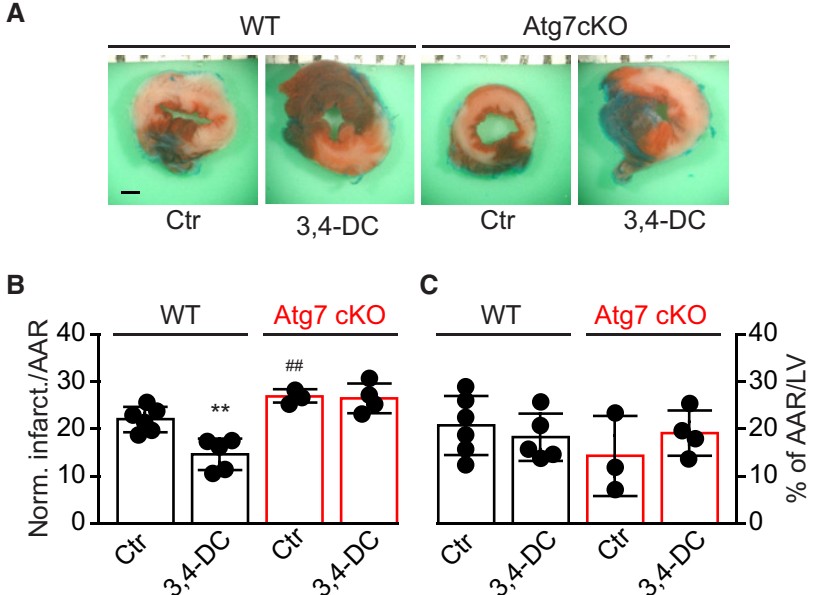

**Figure 7. 3,4-DC confers autophagy-dependent cardioprotection from ischemic injury.**

A  Twelve-week-old wild-type and cardiac-specific Atg7 knockout (Atg7cKO) mice were injected with corn oil (vehicle control, Ctr) or 3,4-DC 24 h before surgery and subjected to 3 h of prolonged ischemia. Representative images of left ventricular (LV) myocardial sections after alcian blue and triphenyltetrazolium chloride (TTC) staining are depicted. Scale bar equals 1 mm.

B, C  The size of the infarction area (indicated by pale color) per area at risk (AAR) (B) and the global AAR per LV (C) was measured (mean value ± SEM, $n = 3–6$, **$P < 0.01$, ##$P < 0.01$ vs. WT/Ctr; Student's $t$-test).

CRMs such as spermidine and resveratrol reduce the growth of tumors in the context of anthracycline or oxaliplatin (OXA)-based chemotherapies through the stimulation of an anticancer immune response (Michaud *et al*, 2011; Pietrocola *et al*, 2016). 3,4-DC treatment induced autophagic flux in murine MCA205 fibrosarcomas that were further used to establish tumors in syngeneic C57Bl/6 immunocompetent hosts (Fig 8A). In contrast to the chemotherapeutic drug mitoxantrone (MTX), 3,4-DC alone failed to reduce the infiltration of tumor by regulatory T cells (Tregs) and to improve the CD8$^+$/Treg ratio, yet tended to improve these MTX effects (Fig 8B–E). Indeed, the combination of 3,4-DC and MTX was more efficient in reducing tumor growth than MTX alone (Fig 8F), and this combinational effect was lost in the context of a T lymphocyte deficiency (i.e., when MCA205 cancer cells were implanted into athymic *nu/nu* mice (Fig 8G) or when the MCA205 cells were rendered autophagy-deficient due to the knockdown of *Atg5* (Fig 8H and I). The chemotherapy-improving effects of 3,4-DC were also observed in combination with oxaliplatin (Fig EV5A and B) or in TC-1 non-small-cell lung cancers treated with MTX (Fig EV5D and E). 3,4-DC exhibited beneficial effects in combination with OXA or MTX when the agents were administered to immunocompetent mice. Of note, the combination of 3,4-DC with chemotherapy lost its efficacy in immunodeficient *nu/nu* animals (Fig EV5C and F). Moreover, when TFEB and TFE3 were knocked down in the cancer cells, the favorable interaction between MTX and 3,4-DC leading to stronger tumor growth reduction than with MTX alone was lost (Fig 8J–L).

## Discussion

The present study led to the identification of a novel candidate CRM, 3,4-DC. Indeed, 3,4-DC was able to stimulate the CRM-definitory triad of cellular events on multiple human cell lines: (i) deacetylation of cytoplasmic proteins, (ii) induction of autophagy, and (iii) lack of toxicity. Mouse experiments then led to the validation of the 3,4-DC effects *in vivo*, with the demonstration that 3,4-DC induces autophagic flux *in vivo*, in the liver and heart, as indicated by the fact that 3,4-DC stimulated the accumulation of the lipidated form of LC3 (LC3-II) even in conditions in which the last step of autophagy is blocked by leupeptin or chloroquine. Moreover, mouse experiments confirmed that 3,4-DC did not induce major stigmata of toxicity and that it actually mediated cardioprotection and stimulated autophagy-dependent anticancer immune responses in the context of immunogenic chemotherapies. Altogether, these results validate the approach to search for candidate CRMs by means of *in vitro* screening experiments on human cell lines, because the lead compound identified in this screen, 3,4-DC, has desirable pharmacological properties in preclinical models of heart disease and cancer. At this point, however, it remains to be determined whether 3,4-DC is able to increase the health span and lifespan of mammalian species so that it can be considered as a true CRM.

Although 3,4-DC resembles known CRMs with respect to the aforementioned features, it has also some unexpected properties that distinguish it from established CRMs. Similar to prolonged starvation, but at difference with established CRMs such as spermidine and resveratrol, 3,4-DC fully relied on transcription and translation

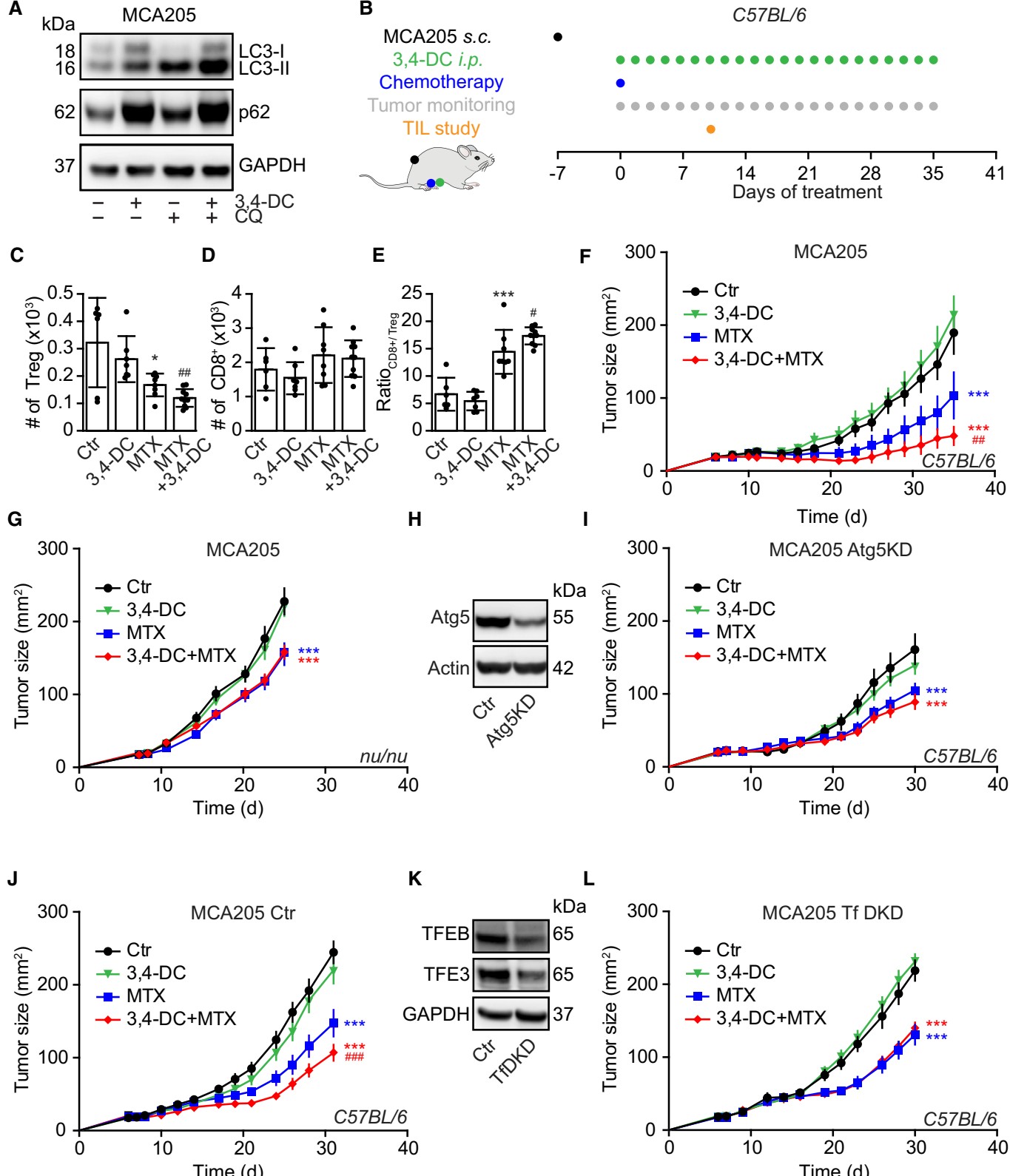

**Figure 8.**

◀

**Figure 8.  3,4-DC improves the efficacy of anticancer chemotherapy.**

A    Induction of autophagy in murine MCA205 fibrosarcomas. Cells were treated with 3,4-DC alone or in combination with chloroquine, and were harvested 6 h later for immunoblot detection of LC3 lipidation.

B    Schematic overview of the *in vivo* treatment of MCA205 fibrosarcomas with mitoxantrone (MTX) and 3,4-DC, alone or in combination.

C–E  Cytofluorometric characterization of tumor-infiltrating lymphocytes (TIL), in particular FOXP3[+] regulatory T cells (Treg), CD8[+] cytotoxic T lymphocytes, and the ratio of CD8[+] T cells over Treg.

F–I  Growth kinetic of MCA205 fibrosarcomas that were either wild-type (F, G) or *Atg5KD* (H, I) and were evolving in immunocompetent C57Bl/6 mice (F, I) or immunodeficient *nu/nu* mice (G), treated as indicated in (B).

J–L  Immunocompetent C57Bl/6 mice were subcutaneously inoculated with TFEB/TFE3 double knockdown MCA205 cells or its scramble control cells (K). When tumors became palpable, mice were treated as indicated in (B). Tumor growth curves from mice subjected to 3,4-DC administration alone or in combination with MTX are shown (J, L).

Data information: Asterisks indicate significant effect of MTX with respect to untreated controls (mean value $\pm$ SEM, *$P$ < 0.05, ***$P$ < 0.001; Student's $t$-test), while hash symbols refer to the comparison of the effects of MTX plus 3,4-DC to MTX alone ([#]$P$ < 0.05, [##]$P$ < 0.01, [###]$P$ < 0.001; Student's $t$-test) ($n$ = 6–14). Samples for immunoblots were run on one gel (A, H) or several parallel gels (K), then blotted, cut into horizontal stripes, and probed separately.

Source data are available online for this figure.

to induce autophagy (Sahani *et al*, 2014). This effect involved the pro-autophagic transcription factors TFEB and TFE3, which translocated from the cytoplasm to the nucleus upon addition of 3,4-DC, both in human cells (*in vitro*) and in mouse hepatocytes or cardiomyocytes (*in vivo*). Depletion or deletion of TFEB and/or TFE3 largely prevented the transcriptional induction of major pro-autophagic genes, reduced lysosomal biogenesis, and abolished the induction of autophagy by 3,4-DC, underscoring the importance of these transcription factors for the 3,4-DC effect. The transcription-dependent autophagy-inducing activity of 3,4-DC clearly differs from that of 4,4′-DC, which does not activate the TFEB/TFE3 pathway and rather inhibits GATA transcription factors. That said, it should be mentioned that 3,4-DC acts at relatively high concentrations (10–30 μM) calling for the search of 3,4-DC analogues or derivatives that have a higher potency.

It will be interesting to further characterize the 3,4-DC-mediated cytoprotective, anticancer, and possible anti-aging effects since they are mechanistically different from other CRMs with respect to TFEB/TFE3 dependence. This might imply that 3,4-DC could be combined with other CRMs to mediate additive or even synergistic effects of autophagy induction and thus disease protection.

# Materials and Methods

## Cell culture and chemicals

Culture media and supplements for cell culture were purchased from Gibco-Life Technologies (Carlsbad, CA, USA) and plasticware from Greiner Bio-One (Kremsmünster, Austria). All cells other than PC12 which were cultured in RPMI-1640 containing 5% fetal bovine serum and 10% horse serum were maintained in Dulbecco's modified Eagle's medium (DMEM) supplemented with 10% fetal bovine serum, 100 units/ml penicillin G sodium, and 100 μg/ml streptomycin sulfate at 37°C under 5% $CO_2$. All chalcones were purchased from Extrasynthese (Genay, France) and LysoTracker Red from Life Technologies; rapamycin, chloroquine, bafilomycin A1, cycloheximide, and actinomycin D were obtained from Sigma-Aldrich (St. Louis, MO, USA).

## High-content microscopy

U2OS cells or H4 cells stably expressing GFP-LC3, RFP-Lamp1, GFP-TFEB, or GFP-RFP-LC3, or PC12 expressing Q74-GFP were seeded in 384-well black microplates and incubated for 24 h. After treatment, cells were fixed with 4% paraformaldehyde (PFA, w/v in PBS) for 20 min at room temperature and stained with 10 μg/ml Hoechst 33342 in PBS. Image acquisition was performed using an ImageXpress Micro XL automated microscope (Molecular Devices, Sunnyvale, CA, US) equipped with a 20 X PlanApo objective (Nikon, Tokyo, Japan). At least four view fields were acquired per well, and experiments involved at least triplicate assessment. Quantitation was usually done on 1,200–2,400 cells per condition. Upon acquisition, images were analyzed using the Custom Module Editor functionality of the MetaXpress software (Molecular Devices). Briefly, cells were segmented and divided into nuclear and cytoplasmic regions based on the nuclear Hoechst staining and cytoplasmic GFP or RFP signal. GFP-LC3 dots were detected using automated thresholding, and their number and surface were measured in the cytoplasmic or/and nuclear compartment. GFP-TFEB intensities were also systematically measured in both compartments. Data processing and statistical analyses were performed using the R software (http://www.r-project.org/).

## *In vivo* experimentation

All mice were maintained in a temperature-controlled and pathogen-free environment with 12-h light/dark cycles, with food and water *ad libitum*. C57BL/6 mice were obtained from RIKEN BioResource Center (Ibaraki, Japan). Twelve-week-old male wild-type and cardiac-specific Atg7 knockout (Atg7cKO) mice were injected with vehicle control or 3,4-DC 24 h before surgery and subjected to 3 h of prolonged ischemia. Myocardial tissue was excised and stained with alcian blue and triphenyltetrazolium chloride (TTC) according to standard procedures. Following this, images of left ventricular (LV) myocardial sections were acquired and the size of the infarction area per global area at risk (AAR) per LV was measured. Alternatively, 8-week-old female GFP-LC3 transgenic (Tg-GFP-LC3) mice, containing a rat LC3-EGFP fusion under the control of the chicken β-actin, were treated with 3,4-DC hearts were thereafter collected and processed for the detection of GFP-LC3. All experiments were approved by the Rutgers-New Jersey Medical School's Institutional Animal Care and Use Committee.

For tumor growth experiments, female wild-type C57BL/6 mice at the age of 6–8 weeks were obtained from Envigo, France (Envigo, Huntingdon, UK), while athymic female nude mice (*nu/nu*) at the

age of 6–8 weeks came from the Gustave Roussy Cancer Center. Animals were kept under controlled conditions at the animal facility at the Gustave Roussy Campus Cancer. Animal experiments were conducted in compliance with the EU Directive 63/2010 and protocols 2019_030_20590 and were approved by the Ethical Committee of the Gustave Roussy Campus Cancer (CEEA IRCIV/IGR no. 26, registered at the French Ministry of Research). MCA205 or TC-1 tumors were established in C57BL/6 hosts by subcutaneously inoculating $5 \times 10^5$ cells. When tumors became palpable, mice were treated with 230 mg/kg 3,4-DC dissolved in corn oil (Sigma-Aldrich) or an equivalent volume of vehicle alone or in combination with 5.17 mg/kg mitoxantrone (MTX, Sigma-Aldrich), or 10 mg/kg oxaliplatin (OXA, Sigma-Aldrich) by intraperitoneal injection. On the following days, mice well-being and tumor growth were monitored and documented. Animals were sacrificed when tumor size reached endpoint or signs of obvious discomfort were observed following the EU Directive 63/2010 and our Ethical Committee advice. Starvation was achieved by removing food for 24 h. Following animals received 40 mg/kg of the autophagic flux inhibitor leupeptin in PBS i.p. for 2 h before the organs were collected for subsequent experiments.

## Detection of protein deacetylation

U2OS-GFP-LC3 stable cells were seeded in 384-well microplates for 24 h. After experimental treatments, cells were fixed with 4% paraformaldehyde for 20 min at room temperature. Thereafter, cells were incubated with an antibody specific for acetyl-alpha-tubulin (#5335, 1:500, Cell Signaling Technology) in 5% bovine serum albumin (BSA, w/v in PBS) for 1 h to block non-specific binding sites and acetylated tubulins, followed by overnight incubation at 4°C with specific antibody to detect acetylated proteins at lysines (#623402, 1:400, BioLegend). After washing several times with PBS, cells were incubated in Alexa Fluor™ conjugates (Life Technologies) against the primary antibody for 2 h at room temperature. Nuclei were stained by incubating the cells with 10 μg/ml Hoechst 33342 in PBS. Fluorescent images were acquired and analyzed as described before.

## Immunoblotting

After treatment, cells were collected and lysed in cold RIPA lysis and extraction buffer (Thermo Fisher, Carlsbad, CA, USA) containing Pierce Protease and Phosphatase Inhibitor Mini Tablet (Thermo Fisher) on ice for 40 min. After centrifugation at 12,000 *g* for 15 min, supernatants were heated in sample buffer (Thermo Fisher) at 100°C for 10 min. Protein samples were separated on pre-cast 4–12% polyacrylamide NuPAGE Bis–Tris gels (Life Technologies) and electro-transferred to PVDF membranes (Millipore Corporation). Membranes were probed overnight at 4°C with primary antibodies specific to LC3 (#2775, 1:1,000, Cell Signaling Technology), p62 (#ab56416, 1:10,000, Abcam), Atg5 (#A2859, 1:1,000, Sigma-Aldrich), GAPDH (#ab8245, 1:10,000, Abcam), Atg14 (#5504, 1:1,000, Cell Signaling Technology), P-p70S6K (#9234, 1:1,000, Cell Signaling Technology), p70S6K (#9202, 1:1,000, Cell Signaling Technology), TFEB (#4240, 1:1,000, Cell Signaling Technology), TFE3 (ab93808, 1:2,000, Abcam), MiTF(#97800, 1:1,000, Cell Signaling Technology), Lamp1 (#15665, 1:1,000, Cell Signaling Technology), H3(#4499, 1:2,000, Cell Signaling Technology), HA (#ROAHAHA,

1:2,000, Sigma-Aldrich), TSC2 (#4308, 1:1,000, Cell Signaling Technology), P-TFEB-S211 (#37681, 1:500, Cell Signaling Technology), GATA2 (#710242, 1:500, Thermo Fisher), and actin (#ab49900, 1:10,000, Abcam) followed by incubation with appropriate horse-radish peroxidase (HRP)-conjugated secondary antibodies (Southern Biotech, 1:5,000, Cambridge, UK). Immunoreactive bands were visualized with ECL prime Western blotting detection reagent (Sigma-Aldrich). Nuclear fractionation was performed with the nuclear extraction kit (ab113474, Abcam) according to the manufacturer's advice, followed by immunoblotting as described above. Usually, blots were cut into horizontal strips to probe for different proteins with different electrophoretic mobility (e.g., LC3, p63, and GAPDH) in parallel. For phosphoproteins, the blots were stripped and reprobed to detect the protein independent from its phosphorylation status. In certain instances, samples were run on parallel gels and were probed separately. Equal loading was controlled for all blots and one representative loading control is depicted, as indicated in the figure legends.

## Immunofluorescence

Cells were fixed with 3.7% paraformaldehyde for 20 min at room temperature, permeabilized with 0.1% Triton on ice, and then blocked with 5% bovine serum albumin (BSA, w/v in PBS) for 1 h, followed by overnight incubation at 4°C with antibodies specific to TFEB (#4240, 1:400, Cell Signaling Technology) or TFE3 (ab93808, 1:200, Abcam). Thereafter, Alexa Fluor™ conjugates (Life Technologies) against the primary antibody were applied for 2 h at room temperature. Cells were then washed and imaged by high-content microscopy as described above.

## RNA interference

Cells were seeded, let adhere for 24 h, and following transfected by the means of DharmaFECT™ transfection reagents (GE Healthcare Dharmacon, Chalfont St Giles, UK), according to the manufacturer's advice. Transfection efficacy was routinely monitored 48 h post-transfection. Specific siRNA sequences (GE Healthcare Dharmacon) were used as detailed below: siCtr (5′-UAGCGACUAAACACAUCAA-3′), siTFEB-#1 (5′-CUACAUCAAUCCUGAAAUG-3′), siTFEB-#2 (5′-AGACGAAGGUUCAACAUCA-3′), siTFEB-#3 (5′-CGGGAGUACCUGUCCGAGA-3′), siTFE3-#1 (5′-GGAAUCUGCUUGAUGUGUA-3′), siTFE3-#2 (5′-GCUCAAGCCUCCCAAUAUC-3′), siTFE3-#3 (5′-CGCAGGCGAUUCAACAUUA-3′), siTFE3-#4 (5′-CAGAGCAGCUGGACAUUGA-3′), siMITF-#1 (5′-UGGCUAUGCUUACGCUAAA-3′), siMITF-#2 (5′-AGAACUAGGUACUUUGAUU-3′), siMITF-#3 (5′-AGACGGAGCACACUUGUUA-3′), siMITF-#4 (5′-GAACACACAUUCACGAGCG-3′), siGATA2-1# (5′-UCGAGGAGCUGUCAAAGUG-3′), siGATA2-2# (5′-GAAGAGCCGGCACCUGUUG-3′), siLamp1-1#(5′-CAACAGAGUAACAUCGAA-3′), and siLamp1-2# (5′-GGCUUAGGGUCCUGUCGAA-3′).

## Transcription factor knockout or knockdown

TFEB and TFE3 knockout cells were generated with CRISPR-Cas9 plasmid U6gRNA-Cas9-2A-RFP targeting TFEB sequence (5′-AGTCGCCACCACCTGTGCCTGG-3′) or TFE3 sequence (5′-TCATGCGGCCGAACCAGCTCGG-3′). TFEB and TFE3 double stable knockdown was achieved by transfecting the cells with the pSMART mCMV vector expressing shRNA targeting TFEB (5′-GCAGCCACCT

**The paper explained**

**Problem**
Caloric restriction (CR) can extend health span and extend longevity. This effect has been broadly characterized on many different species. Given the logistic difficulties to maintain CR for longer periods, the concept of "CR mimetics" (CRMs) has been developed. CRMs evoke all beneficial health effects of CR without the need for major changes in lifestyle.

**Results**
Herein, we designed a systems biology approach for the discovery of novel candidate CRMs. 3,4-dimethoxychalcone (3,4-DC) was identified to induce global protein deacetylation and stimulated autophagic flux. Mechanistically 3,4-DC depended on transcription factor EB (TFEB) and E3 (TFE3)-regulated transcription to trigger autophagy. 3,4-DC-mediated autophagy-dependent cardioprotective effects and improved the efficacy of anticancer chemotherapy *in vivo*.

**Impact**
3,4-DC may prevent the development of disease and might increase the clinical efficacy of anticancer treatment through the induction of autophagy.

GAACGTGTA-3′) and the pLKO.1-CMV-neo vector expressing shRNA targeting TFE3 (5′-TGTGGATTACATCCGCAAATT-3′).

## Autophagy measurement on tissue sections

Liver and heart tissues were fixed with 10% neutral buffered formalin at room temperature for 4 h and then transferred into 30% sucrose diluted in PBS for 24 h at 4°C. Thereafter, the organs were embedded in O.C.T solution, and consecutive tissue sections were cut by means of a cryostat. Samples were stained with DAPI to detect nuclei, and images were acquired by confocal microscopy. GFP-LC3 dot area was measured by ImageJ.

## Enucleation experiment

U2OS-GFP-LC3 cells were suspended in 3 ml of complete medium containing 10 μg/ml cytochalasin B and incubated for 45 min at 37°C. This cell suspension was then gently transferred onto an already prepared discontinuous Ficoll-Paque (GE Healthcare) density gradient solutions (3 ml 55%, 1 ml 90%, and 3 ml 100% Ficoll-Paques supplemented with complete medium), which had been confected in ultracentrifuge tubes and pre-equilibrated in a $CO_2$ incubator set at 37°C overnight. The gradient solution with cells was centrifuged in a prewarmed rotor SW41 (Beckman) for 20 min at 77,000 $g$, 29°C. The cytoplasts were enriched in the 90% Ficoll-Paque layer.

## Real-time PCR

mRNA was extracted with the RNA Purification Kit (K0731, Life Technologies), which was then transformed into cDNA with the cDNA Synthesis Kit (K1641, Life Technologies). mRNA level was detected and quantified by the Power SYBR Green PCR Kit (4368706, Life Technologies) all using the manufacturers' recommendations. RT–PCR primers are listed in Appendix Table S3.

## PI staining

Cells were treated as indicated, and then trypsinized and incubated in 200 μl complete medium supplemented with 1 μg/ml propidium iodide (PI) for 20 min at 37°C. Following this, the cells were analyzed by flow cytometry.

## Statistical analysis

Unless otherwise mentioned, data are reported as means ± SD of triplicate determinations and experiments were repeated at least twice yielding similar results, and statistical significance was assessed by Student's $t$-test. $P$-values are provided in Appendix Table S4. TumGrowth and GraphPad were used to analyze *in vivo* data arising from murine models (Enot *et al*, 2018). Statistics and number of animals used are provided in Appendix Tables S5 and S6.

**Expanded View** for this article is available online.

## Acknowledgements

GK is supported by the Ligue contre le Cancer (équipe labelisée); Agence National de la Recherche (ANR)—Projets blancs; ANR under the frame of E-Rare-2, the ERA-Net for Research on Rare Diseases; Association pour la recherche sur le cancer (ARC); Cancéropôle Ile-de-France; Institut National du Cancer (INCa); Fondation de France; Fondation pour la Recherche Médicale (FRM); the European Commission (ArtForce); the European Research Council (ERC); Institut Mérieux, the LabEx Immuno-Oncology; the SIRIC Stratified Oncology Cell DNA Repair and Tumor Immune Elimination (SOCRATE); the SIRIC Cancer Research and Personalized Medicine (CARPEM); and the Paris Alliance of Cancer Research Institutes (PACRI). FM is grateful to the Austrian Science Fund FWF (Austria) for grants, P29262, P24381, P29203, P27893, and "SFB Lipotox" (F3012), as well as to Bundesministerium für Wissenschaft, Forschung und Wirtschaft and the Karl-Franzens University for grant "Unkonventionelle Forschung" and grant DKplus Metabolic and Cardiovascular Diseases (W1226). We acknowledge support from NAWI Graz and the BioTechMed-Graz flagship project "EPIAge". LD is grateful to the Erasmus+ International mobility program for financial support.

## Author contributions

GC, WX, JN, PL, VS, TP, JT, MB, SJH, LD, SL, MM, and NT conducted experiments and analyzed data AS, FP, DC-G, AZ, JS, and FM assisted with preparation of chemical compound library, data analysis, and interpretation, and OK and GK generated the figures and wrote the article.

## Conflicts of interest

D.C.-G., F.M., O.K. and G.K. are co-founders of Samsara Therapeutics.

## For more information

www.kroemerlab.com;
https://github.com/kroemerlab

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
