## [Review Process File · EMBO Molecular Medicine]

3,4-Dimethoxychalcone induces autophagy through activation of the transcription factors TFE3 and TFEB

Guo Chen, Wei Xie, Jihoon Nah, Allan Sauvat, Peng Liu, Federico Pietrocola, Valentina Sica, Didac Carmona-Gutierrez, Andreas Zimmermann, Tobias Pendl, Jelena Tadic, Martina Bergmann, Sebastian J. Hofer, Lana Domuz, Sylvie Lachkar, Maria Markaki, Nektarios Tavernarakis, Junichi Sadoshima⁸, Frank Madeo, Oliver Kepp & Guido Kroemer

Review timeline:

Submission date:	13 February 2019
Editorial Decision:	2 April 2019
Revision received:	16 July 2019
Editorial Decision:	9 August 2019
Revision received:	3 September 2019
Accepted:	16 September 2019

Editor: Céline Carret

Transaction Report:

1st Editorial Decision

2 April 2019

Thank you for the submission of your manuscript to EMBO Molecular Medicine. We have now heard back from the three referees whom we asked to evaluate your manuscript.

While the referees find the data to be of interest, still quite a few issues were raised that would require your attention. You will see that the lifespan extension experiment was criticized and referee 3 recommends extending the data to yeast and / or worms, as the effects are small, confirmation should come from another source. Referee 1 requests additional details regarding dose and sex-dependencies on lifespan. Referee 2 requests streamlining and better structuring of the paper. Ref 3 questions whether the compound is a real CRM, and given a recent paper from same authors (PMID: 30783116), questions the novelty of the findings. To address this limitation, we would encourage you to provide more convincing mechanistic data on how the compound induces autophagy.

We would welcome the submission of a revised version within three months for further consideration and would like to encourage you to address all the criticisms raised as suggested to improve conclusiveness and clarity. Please note that EMBO Molecular Medicine strongly supports a single round of revision and that, as acceptance or rejection of the manuscript will depend on another round of review, your responses should be as complete as possible.

I look forward to receiving your revised manuscript.

***** Reviewer's comments *****

Referee #1 (Remarks for Author):

Manuscript Number: EMM-2019-10469

Title: 3,4-Dimethoxychalcone induces autophagy through activation of transcription factors TFE3 and TFEB

Corresponding Author: Dr. Keep

Calorie restriction (CR) is the only known established intervention that has consistently been shown to enhance healthspan and lifespan in various model organisms. Searching for small molecules that mimics the beneficial effect of CR has been a hot area of research in the past decade, as they may offer more pragmatic alternatives to CR to promote health and longevity as well as to help elucidate the underlying mechanisms of CR. In this paper, Chen et al. screened a collection of 200 polyphenols and polyamines for their capacity to induce autophagosome formation (GFP-LC3) without apparent toxicity in human osteosarcoma U2OS cells. The resulting compound, 3,4-dimethoxychalcone (3,4-DC), was subjected to extensive characterization in its effects on cytoplasmic protein acetylation, the S6K signaling pathway, autophagy process, lifespan extension in flies as well as on cytoprotecting and antitumor treatment in in vitro models. Furthermore, the authors identified novel transcriptional regulations of autophagy elicited by 3,4-DC. Relevant experimental models were used, and the data presented is convincing. However, a key aspect of the CR mimetic, lifespan extension, was under studied. First, 3,4-DC only extended the mean lifespan by less than 10% in flies. Genetic inhibition of the mTOR-S6K pathway leads to 20-40% mean lifespan extension in male flies (Kapahi, 2004). This should be at least acknowledged. Spermidine and Rapamycin, two proposed CRMs, cause similar effects on fly lifespan. Second, it is not clear which sex was used in the lifespan experiment and only one concentration of 3,4-DC was used. Considering the CRM-like effects of 3,4-DC on S6K pathway, protein acetylation, and autophagy, its limited effects on lifespan is surprising and it would be informative if the authors explained what potentially may account for this. CRMs are known to exert their beneficial effects in a sex and dose-dependent manner. The authors could address at least sex specific effects further.

Referee #2 (Remarks for Author):

While experiments to validate the effects of 3,4-DC across different cell types and models is valuable and novel, in general the manuscript is difficult to read due to the switching back and forth between different cell lines and model organisms, from humans to mice to flies to yeast, and ranging from cardioprotective effects to anticancer effects. It might be useful to streamline the manuscript. There are some aspects related to the micromolar doses required for these compounds to exert the biological effects concluded that need to be addressed in the manuscript. These CRMs are likely to have pleiotropic effects through many different direct targets and it is likely that the deconvolution would be a difficult task, this should be reflected in the manuscript.

Additional comments:

1. In Fig 4G-K, while CHX is shown to decrease 3,4-DC effects on lysosomal dots, the data could be strengthened by showing the effects of CHX on Lamp1 mRNA and protein levels.
2. Does suppression of Lamp1 by siRNA also blunt downstream effects of 3,4-DC?
3. What are the effects of the individual knockdown of TFE3 on Lamp1 and other autophagy-related genes/proteins?
4. It is unclear why 3,4-DC reduces HA-p62 in Fig S1C-D at 16 h, whereas it increases p62 at the longer time points in other experiments throughout the manuscript, example Fig 2B.
5. In the conclusion, the authors claim that 3,4-DC stimulates deacetylation of cytoplasmic proteins, however, the only piece of data that suggests this in the manuscript is Fig S1A-B which is looking at nuclear histones.
6. The use of BafA1 while present in a number of figures is not explained in the text.

Referee #3 (Remarks for Author):

This manuscript presents data arguing that the chalcone 3,4-Dimethoxychalcone induces autophagy, and does so via the transcription factors TFE3 and TFEB. I found some of the supporting data convincing, but the manuscript could be strengthened in certain aspects. Overall I like the study, but their recently published Nature Communications paper on another autophagy-inducing chalcone (not cited in this paper) diminishes the novelty of this one. The concept has been established now. In addition, two of their *in vivo* phenotypes are problematic. The effects on mean fly lifespan are extremely small, and the effects they observed in their cardiac ischemia model do not seem to be statistically significant.

Specific comments.

In the introduction, a few of the references should be altered.

In this sentence, reference for worms should be Hansen et al. Plos Genet, 2008.

"Direct genetic interventions to inhibit autophagy prevent lifespan extension by CR in yeast and nematodes (Eisenberg, Knauer et al., 2009, Morselli, Maiuri et al., 2010)."

Incorrect reference here: Conversely, genetic induction of autophagy by removal of the autophagy inhibitor mTOR can induce a longevity phenotype in nematodes and flies (de Cabo, Carmona-Gutierrez et al., 2014, Kapahi, Zid et al., 2004). The nematode reference should be: Vellai et al, Nature, 2003.

The idea of interventions that mimic caloric restriction has been around for a long time (see papers and book by L. Guarente, for example), so this is not referenced correctly. That said, it's not clear to me why this compound is considered a caloric restriction mimetic. Simply inducing autophagy and changes in protein acetylation does not seem sufficient, and could be confusing to the reader. For example, the authors argue that the compound can produce effects in the animal at concentrations that do not induce weight loss. So it doesn't mimic CR in that sense. Also the tiny (if real) life extension they observe in flies is nothing like the extension produced by caloric restriction. It would be more helpful if the authors discuss some of the similarities they observed between the effect of this compound and CR without giving it the label "CRM".

Page 5: In this sentence, I think the authors might mean p62 levels, not lipidation. Please clarify. "3,4-DC caused a dose dependent increase in LC3 lipidation (detectable as an increase in the electrophoretic mobility of LC3 yielding LC3-II) as well as that of sequestosome-1 (SQSTM1, best known as p62) (Fig. 2A,C)."

In this sentence: "...neuroglioma H4 cells (D,G) stably expressing GFP-LC3 were treated with a library of chalcones (30 μ M). Change "G" to "H", right?

Figure 1, legend. Need a better assay for cell viability, such as a propidium iodide uptake. Was such an assay performed subsequently (in follow-up studies) on cells treated with the compound?

Figure 2, G-J and Figure 3G. How many cells were scored?

Figure 2K. The *Drosophila* lifespan data are inadequate. From the data given here, the magnitude of the lifespan extension is too small to be accepted as fact. How many times was the experiment performed? (They should show data in a table for all repetitions.) Where are the data points in the graph? How many flies were observed? What is the magnitude of the lifespan increase? Was it tried in yeast? Worms? I recommend that this conclusion either be strengthened by additional or experiments or omitted. In the abstract and text, the authors should refer to a "modest increase in mean lifespan", "rather than "increase in lifespan..." but only if it's reproducible.

Fig 2 legend, change "M" to "L". (Right?)

Page 6, The reduction in TOR activity could potentially cause the increase in TFEB activity. Is this mentioned anywhere? More generally, how do the authors propose that the chalcone activates autophagy?

Page 8, mouse experiments. What was the dose of the chalcone administered to the mice?

In general it appears that the standard error of the mean was calculated for the various experiments. The important parameter in most cases is the standard deviation. Please show SD instead of SEM on the graphs.

Figure 7, What is the P value? Are the results statistically significant? If they are not statistically significant the experiment should not be included in the paper.

Need background and refs describing the justification for testing this compound in models of cancer immunology. Autophagy is known to promote cancer cell growth, for example.

Is Figure 8 C-E really meaningful?

Fig S6, Title does not relate to fig "3,4-DC induces autophagy in vivo." But it's about cancer.

The authors state that there is no conflict of interest. Do they have any involvement with biotech companies trying to activate autophagy?

1st Revision - authors' response

16 July 2019

Referee #1

General critique by the reviewer:

Calorie restriction (CR) is the only known established intervention that has consistently been shown to enhance healthspan and lifespan in various model organisms. Searching for small molecules that mimics the beneficial effect of CR has been a hot area of research in the past decade, as they may offer more pragmatic alternatives to CR to promote health and longevity as well as to help elucidate the underlying mechanisms of CR. In this paper, Chen et al. screened a collection of 200 polyphenols and polyamines for their capacity to induce autophagosome formation (GFP-LC3) without apparent toxicity in human osteosarcoma U2OS cells. The resulting compound, 3,4-dimethoxychalcone (3,4-DC), was subjected to extensive characterization in its effects on cytoplasmic protein acetylation, the S6K signaling pathway, autophagy process, lifespan extension in flies as well as on cytoprotecting and antitumor treatment in in vitro models. Furthermore, the authors identified novel transcriptional regulations of autophagy elicited by 3,4-DC. Relevant experimental models were used, and the data presented is convincing.

Our response:

We thank the reviewer for accurately summarizing our work and her/his generally positive evaluation.

Specific critique by the reviewer

However, a key aspect of the CR mimetic, lifespan extension, was under studied. First, 3,4-DC only extended the mean lifespan by less than 10% in flies. Genetic inhibition of the mTOR-S6K pathway leads to 20-40% mean lifespan extension in male flies (Kapahi, 2004). This should be at least acknowledged. Spermidine and Rapamycin, two proposed CRMs, cause similar effects on fly lifespan. Second, it is not clear which sex was used in the lifespan experiment and only one concentration of 3,4-DC was used. Considering the CRM-like effects of 3,4-DC on S6K pathway, protein acetylation, and autophagy, its limited effects on lifespan is surprising and it would be informative if the authors explained what potentially may account for this. CRMs are known to exert their beneficial effects in a sex and dose-dependent manner. The authors could address at least sex specific effects further.

Our response:

We thank the reviewer for challenging our fly experiments. Indeed, we performed a total of 8 experiments, finding only in 5 of them a minor (less than 10%) lifespan extension in females after treatment with 3,4-DC. We also tested 3,4-DC on *C. elegans* (in collaboration with

Nektarios Tavernerakis), and we did not find any proof for autophagy induction or any lifespan extension. Finally, in yeast (*S. cerevisiae*), 3,4-DC failed to improve survival in chronological aging experiments. This led us to the conclusion that 3,4-dimethoxychalcone (3,4-DC) does not induce longevity phenotypes in these non-mammalian model organisms. As a result, we deleted the fly lifespan experiment (Fig. 2K in the original submission) from the paper.

Referee #2

General critique by the reviewer:

While experiments to validate the effects of 3,4-DC across different cell types and models is valuable and novel, in general the manuscript is difficult to read due to the switching back and forth between different cell lines and model organisms, from humans to mice to flies to yeast, and ranging from cardioprotective effects to anticancer effects. It might be useful to streamline the manuscript. There are some aspects related to the micromolar doses required for these compounds to exert the biological effects concluded that need to be addressed in the manuscript. These CRMs are likely to have pleiotropic effects through many different direct targets and it is likely that the deconvolution would be a difficult task, this should be reflected in the manuscript.

Our response: We thank the reviewer for this general comment. We attempted to streamline our paper by reducing the amount of information from main figure to supplemental material, whenever appropriate. Moreover, we accentuated the structure of the Results in subchapters. We removed data on yeast and flies, focusing the paper on the mammalian system. We critically mentioned that 3,4-DC acts at 10 to 30 microM, which is relatively high for a pharmacological lead compound. However, we were able to add additional evidence in favor of some sort of molecular ‘specificity’ in the mode of action of 3,4-DC. Indeed, the anticancer effects of 3,4-DC were lost when the two TFE transcription factors TFEB and TFE3 were knocked down in cancer cells and such cells were subsequently inoculated into mice. These results, as well others that were added to the paper in response to the reviewer’s specific critique, support the notion that we were able to deconvolute the mode of action of 3,4-DC.

Specific comment No. 1: In Fig 4G-K, while CHX is shown to decrease 3,4-DC effects on lysosomal dots, the data could be strengthened by showing the effects of CHX on lamp1 mRNA and protein levels.

Our response: As requested by the reviewer, we have evaluated the effects of CHX on LAMP1 protein and *LAMP1* mRNA. The 3,4-DC-induced increase in LAMP1 protein expression was abolished by CHX. However, the 3,4-DC-induced increase in *LAMP1* mRNA was not inhibited by CHX.

Specific comment No. 2: Does suppression of Lamp1 by siRNA also blunt downstream effects of 3,4-DC, as show here:

Our response: Driven by the referee, we knocked down *LAMP1* with two different, non-overlapping siRNAs. Although LAMP1 protein was well depleted, the induction of LC3-II (in parental U2OS cells) and GFP-LC3 dots (in stably GFP-LC3 expressing U2OS cells) by 3,4-DC was not affected by this manipulation.

(A, B) LAMP1 knockdown was achieved by two different siRNAs targeting LAMP1 (siLAMP1-1#, siLAMP1-2#) with scramble siRNA as a control (siCtr), followed by 30 μ M 3,4-DC treatment for 16 h. The cells were collected and lysed for western blot, and then antibodies against LAMP1, LC3, p62, or GAPDH were employed to detect protein levels (A). Or the cells were fixed and GFP-LC3 puncta were quantified in (B). Data are means \pm SD.

Specific comment No. 3: What are the effects of the individual knockdown of TFE3 on Lamp1 and other autophagy-related genes/proteins?

Our response: To respond to this question, we performed TFE3 knockouts with two different guidance RNAs. The CRISP/Cas9-based removal of TFE3 partially reduced the induction of GFP-LC3 dots by 3,4-DC. It also abolished the induction of LAMP1 protein by 3,4-DC. Hence, TFE3 does play a role in the effects of 3,4-DC on autophagy and lysosomal biogenesis.

(A) U2OS cells wild type (WT) and TFE3 knockout cells by two different gRNAs targeting TFE3 (TFE3KO-1, TFE3KO-2) were treated with 30 mM 3,4-DC. Cells were fixed and images were acquired by microscopy. GFP-LC3 dots number was counted. Data are means \pm SD (***) = $p < 0.001$ vs WT/3,4-DC). (B, C) U2OS WT, TFE3KO-1, and TFE3KO-2 cells were treated as in (A), and then cells were collected and processed for western blot. LAMP1, LC3, p62, and GAPDH protein levels were measured with the respective antibodies (B). Bands intensities of LAMP1 and GAPDH were measured and their ratio was calculated in (C). Data are means \pm SD of at least three independent experiments (* = $p < 0.05$, ** = $p < 0.01$, no significance: n.s.).

Specific comment No. 4: It is unclear why 3,4-DC reduces HA-p62 in Fig S1C-D at 16h, whereas it increases p62 at the longer time points in other experiments throughout the manuscript, example Fig 2B.

Our response: In our paper, we included data showing that 3,4-DC promoted endogenous p62 synthesis by stimulating TFEB and TFE3 activity. We believe that this transcription/translation-dependent effect explains why 3,4-DC increases p62 at later time points. P62 is an autophagy receptor that is degraded along with other autophagy substrates when autophagy is activated. This explains why the protein is reduced in abundance after short-term induction of autophagy. To distinguish these effects of endogenous, we artificially expressed HA-tagged p62 in a pcDNA3.1 vector in cells, assuming that expression of HA-p62 driven by the CMV promoter should be independent from regulation of gene transcription by TFEB and TFE3. We found that the abundance of this HA-tagged p62 decreased after treatment with 3,4-DC and that this decrease was abolished by the autophagy inhibitor chloroquine, supporting the idea that there is indeed an autophagy-dependent turnover of p62 that explains the early drop in the expression of this protein (Extended view 1 C,D).

Specific comment No. 5: In the conclusion, the authors claim that 3,4-DC stimulates deacetylation of cytoplasmic proteins, however, the only piece of data that suggests this in the manuscript is Fig S1A-B, which is looking at nuclear histones.

Our response: We performed further experiments in addition to those already included in the paper. In sum, 3,4-DC induces general deacetylation of cytoplasmic proteins as shown in Fig 1C, E, in which acetylated protein intensity was measured by means of an anti-lysine antibody staining. Branched-chain amino acids such as L-leucine (Leu) and branched-chain α -ketoacids such as α -ketoisocaproic acid (KIC) undergo oxidative decarboxylation to yield acetyl coenzyme A (AcCoA), while dichloroacetate (DCA) is a pharmacological inhibitor of pyruvate dehydrogenase kinase (PDK) which negatively regulates AcCoA synthesis. These three agents (Leu, KIC and DCA) effectively inhibited 3,4-DC induced deacetylation of cytoplasmic proteins as they suppressed 3,4-DC induced autophagy (new Fig. 1 J,K).

Specific comment No. 6: The use of BafA1 while present in a number of figures is not explained in the text.

Our response: We added a phrase explaining the use of BafA1 to the text.

Referee #3

General critique by the reviewer:

This manuscript presents data arguing that the chalcone 3,4-Dimethoxychalcone induces autophagy, and does so via the transcription factors TFE3 and TFEB. I found some of the supporting data convincing, but the manuscript could be strengthened in certain aspects. Overall I like the study, but their recently published Nature Communications paper on another autophagy-inducing chalcone (not cited in this paper) diminishes the novelty of this one. The concept has been established now. In addition, two of their in vivo phenotypes are problematic. The effects on mean fly lifespan are extremely small, and the effects they observed in their cardiac ischemia model do not seem to be statistically significant.

Our response: We have cited the paper in Nature Communication (that deals with another chalcone, 4,4'-DC) and performed a direct comparison of the pro-autophagic mode of action of 3,4-DC and 4,4'-DC that has been included in the paper. We found that 3,4-DC causes classical autophagy (with autophagosomes in the cytoplasm), while 4,4'-DC causes accumulation of GFP-LC3 dots in or around nuclei. More importantly, 3,4-DC induces autophagy through the activation of transcription factors of the TFE family (in particular TFEB and TFE3), while 4,4'-DC induces autophagy via the inhibition of GATA transcription factors. Finally, we chose more representative pictures to illustrate the effects on the cardiac ischemia model. The statistics have been calculated and reveal a significant cardioprotective effect for 3,4-DC in autophagy-competent wild type (WT) mice, but not in mice in which the essential autophagy gene *Atg7* has conditionally been knocked out in cardiomyocytes.

Specific comment No. 1: In the introduction, a few of the references should be altered. In this sentence, reference for worms should be Hansen et al. Plos Genet, 2008.

“Direct genetic interventions to inhibit autophagy prevent lifespan extension by CR in yeast and nematodes (Eisenberg, Knauer et al., 2009, Morselli, Maiuri et al., 2010).”

Incorrect reference here: Conversely, genetic induction of autophagy by removal of autophagy inhibitor mTOR can induce a longevity phenotype in nematodes and flies (de Cabo, Carmona-Gutierrez et al., 2014, Kapahi, Zid et al., 2004). The nematode reference should be: Vellai et al, Nature, 2003.

Our response: The requested references have been added to the text.

Specific comment No. 2: The idea of interventions that mimic caloric restriction has been around for a long time (see papers and book by L. Guarente, for example), so this is not referenced correctly. That said, it's not clear to me why this compound is considered a caloric restriction mimetic. Simply inducing autophagy and changes in protein acetylation does not seem sufficient, and could be confusing to the reader. For example, the authors argue that the compound can produce effects in the animal at concentrations that do not induce weight loss. So it doesn't mimic CR in that sense. Also the tiny (if real) life extension they observe in flies is nothing like the extension produced by caloric restriction. It would be more helpful if the authors discuss some of the similarities they observed between the effect of this compound and CR without giving it the label “CRM”.

Our response: We have re-discussed this aspect in the revised version of the paper. Our assumption is that caloric restriction mimetics (CRMs) induce autophagy through biochemical pathways that mimic those induced by starvation including the reduction of protein acetylation. The in vitro screen that led us to the identification of 3,4-DC as a potentially interesting pharmacological lead was based on these characteristics: protein deacetylation and autophagy induction. We removed the fly aging experiments from the paper. We did observe that 3,4-DC had some positive effects that are shared with established CRMs, in particular with respect to cardioprotection and anticancer effects. As the reviewer recommended, we tuned down the conclusion that 3,4-DC is a CRM, rather classifying it as a “candidate-CRM”. We clearly mentioned in the discussion that “it remains to be determined whether 3,4-DC is able to increase the healthspan and lifespan of mammalian species so that it can be considered as a true CRM.”

Specific comment No. 3: Page 5: In this sentence, I think the authors might mean p62 level, not lipidation. Please clarify. “3,4-DC caused a dose dependent increase in LC3 lipidation (detectable as an increase in the electrophoretic mobility of LC3 yielding LC3-II) as well as that of sequestosome-1 (SQSTM1, best known as p62)(Fig.2A, C).”

Our response: We corrected the criticized phrase to make sur that we talk about an augmentation in p62 expression, not lipidation.

Specific comment No. 4: In this sentence: “...neuroglioma H4 cells (D, G) stably expressing GFP-LC3 were treated with a library of chalcones (30 μ M).” Change “G” to “H”, right?

Our response: This reviewer is right. This has been corrected.

Specific comment No. 5: Figure 1, legend. Need a better assay for cell viability, such as a propidium iodide uptake. Was such an assay performed subsequently (in follow-up studies) on cells treated with the compound?

Our response: We measured viability with propidium iodine, finding that 3,4-DC did not compromise the viability of the cells. These data have been added to the revised version of the paper.

Specific comment No. 6: Figure 2, G-J and Figure 3G. How many cells were scored?

Our response: For each measurement, we determined the number of fluorescent dots per cell within 4 view fields (20 x objective, usually 100 tp 150 cells per view field) for each well, and measurement in at least triplicate assessments, meaning that each quantification was done in

average on 1200 to 2400 cells per condition. This has been stated in the Materials and Methods.

Specific comment No. 7: Figure 2K. The *Drosophila* lifespan data are inadequate. From the data given here, the magnitude of the lifespan extension is too small to be accepted as fact. How many times was the experiment performed? (They should show data in a table for all repetitions.) Where are the data points in the graph? How many flies were observed? What is the magnitude of the lifespan increase? Was it tried in yeast? Worms? I recommend that this conclusion either be strengthened by additional or experiments or omitted. In the abstract and text, the authors should refer to a “modest increase in mean lifespan”, rather than “increase in lifespan...” but only if it’s reproducible.

Our response: We followed the reviewer’s suggestion to omit the fly experiments because the results are admittedly weak. We did try to recapitulate these results in larger dose-finding experiments in flies, as well as in nematodes and yeast, without finding any major effects in these non-mammalian model organisms.

Specific comment No. 8: Fig2 legend, change “M” to “L”. (Right?)

Our response: The legend has been corrected.

Specific comment No. 9: Page 6, The reduction in TOR activity could potentially cause the increase in TFEB activity. Is this mentioned anywhere? More generally, how do the authors propose that the chalcone activates autophagy?

Our response: According to the literature, TFEB is phosphorylated at residues S142, S138, S211 by mTOR to regulate its activity. Phosphorylation of TFEB at S211 enhances the binding of TFEB to 14-3-3 protein, thus favoring the retention of TFEB in the cytoplasm. Similar to torin, 3,4-DC caused the dephosphorylation of TFEB at S211. These new data have been added to the paper. As shown by immunofluorescence staining and subcellular fractionation followed by immunoblot, the 3,4-DC induced TFEB translocation was also inhibited by TSC2 knockout, a manipulation that causes constitutive activation of mTOR. More importantly, the TSC2 knockout abolished the 3,4-DC-induced induction of p62 and LC3-II protein levels.

Specific comment No. 10: Page 8, mouse experiments. What was the dose of the chalcone administered to the mice?

Our response: The dose administered to mice was specified in the Materials and Methods: 230 mg/kg.

Specific comment No. 11: In general, it appears that the standard error of the mean was calculated for the various experiments. The important parameter in most cases is the standard deviation. Please show SD instead of SEM on the graphs.

Our response: In general, for the *in vitro* experiments standard deviations were calculated. Only for *in vivo* experiments, in mice, standard errors of the mean were calculated. This has been mentioned in the Statistics section of the Materials and Methods, as well as in the Figure legends.

Specific comment No. 12: Figure 7, what is the P value? Are the results statistically significant? If they are not statistically significant the experiments should not be included in the paper.

Our response: The statistics have been calculated and reveal a significant cardioprotective effect for 3,4-DC in autophagy-competent wild type (WT) mice, but not in mice in which the essential autophagy gene *Atg7* has conditionally been knocked out in cardiomyocytes.

Specific comment No. 13: Need background and refs describing the justification for testing this compound in models of cancer immunology. Autophagy is known to promote cancer cell growth, for example.

Our response: We have expanded the justification of the testing of autophagy inducers in tumor immunosurveillance.

Specific comment No. 14: Is Figure 8 C-E really meaningful?

Our response: We opted to keep these data in the principal Fig. 8. Indeed, it appears important the 3,4-DC (which can inhibit mTOR) is not acting as an immunosuppressor in vivo and rather enhances the ratio of immune effectors over immunosuppressive cells in cancer treated with chemotherapy.

However, we added additional results showing that, if combined with chemotherapy, 3,4-DC causes tumor growth reduction only in wild type tumors, not in tumors that lack TFEB and TFE3. This experiment supports the central message of the paper: 3,4-DC acts through the activation of the TFE transcription factors TFEB and TFE3.

Specific comment No. 15: Fig S6, Title does not relate to fig “3,4-DC induces autophagy in vivo.” But it’s about cancer.

Our response: We corrected the title of the legend to the supplemental figure.

Specific comment No. 16: The authors state that there is no conflict of interest. Do they have any involvement with biotech companies trying to activate autophagy?

Our response: After submission of the paper, we got involved in the process of funding a biotech company. This has been mentioned in the conflict-of-interest statement of the revised paper. In addition, the last author of the paper has acquired yet another affiliation that has been added to the article.

2nd Editorial Decision

9 August 2019

Thank you for the submission of your revised manuscript to EMBO Molecular Medicine. We have now received the enclosed reports from the referees that were asked to re-assess it. As you will see the reviewers are now supportive and I am pleased to inform you that we will be able to accept your manuscript pending minor editorial amendments.

I look forward to reading a new revised version of your manuscript as soon as possible, within two weeks.

***** Reviewer's comments *****

Referee #1 (Remarks for Author):

The authors have now addressed my concerns related to the previous version of the manuscript

Referee #2 (Comments on Novelty/Model System for Author):

The new revision has addressed the previous concerns-

Referee #2 (Remarks for Author):

The new submission has addressed the previous critiques raised during the review-

Corresponding Author Name: Oliver Kepp

Manuscript Number: EMM-2019-10469-V2